# *Gains*: Fine-grained Federated Domain Adaptation in Open Set

**Zhengyi Zhong[1,3,*], Wenzheng Jiang[1,*], Weidong Bao[1], Ji Wang[1,†], Qi Wang[2],**
**Guanbo Wang[3], Yongheng Deng[3], Ju Ren[3]**
[1]Laboratory for Big Data and Decision, National University of Defense Technology
[2]College of Science, National University of Defense Technology
[3]Department of Computer Science and Technology, Tsinghua University

## Abstract

Conventional federated learning (FL) assumes a closed world with a fixed total number of clients. In contrast, new clients continuously join the FL process in real-world scenarios, introducing new knowledge. This raises two critical demands: detecting new knowledge, *i.e., knowledge discovery*, and integrating it into the global model, *i.e., knowledge adaptation*. Existing research focuses on coarse-grained knowledge discovery, and often sacrifices source domain performance and adaptation efficiency. To this end, we propose a fine-grained federated domain adaptation approach in open set (***Gains***). *Gains* splits the model into an encoder and a classifier, empirically revealing features extracted by the encoder are sensitive to domain shifts while classifier parameters are sensitive to class increments. Based on this, we develop fine-grained knowledge discovery and contribution-driven aggregation techniques to identify and incorporate new knowledge. Additionally, an anti-forgetting mechanism is designed to preserve source domain performance, ensuring balanced adaptation. Experimental results on multi-domain datasets across three typical data-shift scenarios demonstrate that *Gains* significantly outperforms other baselines in performance for both source-domain and target-domain clients. Code is available at: https://github.com/Zhong-Zhengyi/Gains.

## 1 Introduction

As a typical distributed intelligent model training paradigm, federated learning (FL) [33, 11, 36, 67] has garnered significant attention from researchers in recent years [15, 32, 39, 35, 45, 12, 66, 59, 19, 48]. Conventional FL is often studied in a setup with a fixed number of clients [33, 30], which limits its applicability in a more realistic scenario when new clients, *i.e.*, *target domain*, are allowed to join the learning process. To scale FL effectively in such scenarios, we have to deal with heterogeneous or evolving client data distributions, *e.g.*, IoT networks, cross-device applications. This prompts researchers to prioritize the study of two critical techniques in the field: **(i)** assessing whether the new client contributes previously unseen knowledge [37], referred to as *knowledge discovery*; **(ii)** devising strategies to integrate it into the global model for improving generalization under the updated domain setting [18, 7], which we call *knowledge adaptation*.

**Existing challenges:** Though the practical demands and corresponding techniques are well specified, bottlenecks still remain in achieving the deployment purpose (Fig. 1). Regarding *knowledge discovery*, it is rarely investigated in FL, and existing strategies hardly process complicated scenarios. Take the latest work, FOSDA [37], as an example; it facilitates the discovery of new classes, *i.e.*, *class*

---

[1]* Equal Contribution (zhongzhengyi20@nudt.edu.cn, jiangwenzheng@nudt.edu.cn)

[2]† Corresponding Author (wangji@nudt.edu.cn)

*increment*, in the presence of an open set. However, when faced with domain increment, which is more universal in life, FOSDA encounters the failure of dealing with new domain knowledge. Hence, a more ***fine-grained knowledge discovery*** approach is required to discriminate *class increment* or *domain increment*. As for *knowledge adaptation*, current methods primarily attempt to improve the performance of the newly trained model on target domains. Technically, they often suffer from performance degradation on the source domain while easily overlooking the efficiency of knowledge adaptation [18]. Consequently, we need to introduce a mechanism for ***rapid and balanced knowledge adaptation***, securing seamless integration of new knowledge while consolidating original capabilities.

**Proposed solution:** To this end, this paper presents a *fine-**G**rained feder**a**ted doma**i**n adaptatio**n** method in open **s**et* (***Gains***), which aims at achieving fine-grained knowledge discovery and rapid adaptation without sacrificing the performance on the source domain. Specifically, we discover new knowledge and identify its type (*domain increment* or *class increment*) from the changes in model parameters and extracted features. Then, the federated aggregation process is optimized with

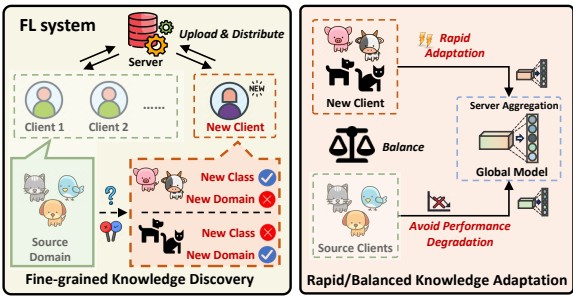

Figure 1: Challenge discription.

the guidance of the quantified parameter and feature's contributions to the target domain, thereby accelerating the integration of new knowledge into the global model. Meanwhile, an ***anti-forgetting mechanism*** (AFM) is designed and used in the training process of the source-domain clients to circumvent source-domain performance degradation, achieving a balance between the target and source domains. To sum up, this work's contribution is three-fold:

- *Adaptation pipeline.* We propose a novel training pipeline within FL that supports fine-grained discovery and discrimination of new knowledge from client updates and efficient integration of incremental knowledge into the global model.

- *Practical solution.* We present an efficient federated optimization method that enables contribution evaluation of diverse components during knowledge adaptation and suppresses performance decline on the source domain.

- *Experimental validation.* We conduct extensive experiments on typical multi-domain datasets under various levels of knowledge shifts. Empirically, *Gains* achieves superior performance on both target and source domain clients over other state-of-the-art methods.

## 2 Related work

**Domain adaptation.** Domain adaptation (DA) can be categorized based on the labeling status of the target domain into unsupervised DA, semi-supervised DA, and supervised DA [47]. They can also be divided based on whether the source domain data is involved into source-dependent DA and source-free DA [24]. The distribution shift is a lasting challenge [52], and typical DA approaches include adversarial learning-based methods and alignment-based methods. Adversarial learning-based methods introduce adversarial networks (such as GANs) to align the feature distributions between the source and target domains [8, 20, 4, 64, 13, 49]. Alignment-based methods achieve alignment between the source and target domains by minimizing the differences in feature or data distributions [23, 10, 54, 9]. Common alignment metrics include KL divergence [34], Maximum Mean Discrepancy (MMD) [25], and Wasserstein distance [34]. In addition, other methods such as self-training [38, 46] and meta-learning [21, 44, 50, 51] have also been applied in DA. Unlike most DA work that considers adapting the source model to the new domain and continual learning that considers catastrophic forgetting [29, 55, 56, 68, 65, 63], we focus on solving the problem of better adaptation to the new domain while avoiding performance degradation in the source domain.

**Federated domain adaptation.** The FDA methods primarily include domain alignment-based, data-based, learning-based, and aggregation optimization-based approaches [31]. Among them, domain alignment consists of feature [53, 7] and gradient alignment [58, 17]. Besides, mixed training approaches are also adopted. For instance, [60] uploads prototypes from different domains to the server for fine-tuning. In data adjustment methods, data augmentation [40, 3, 22] and data generation

[14, 27, 57] are commonly used. Chen et al. [3] generated data with other domain styles on a single client through style transfer between clients. In learning-based approaches, common strategies include adding alignment regularization terms [16], representation learning [1, 61], and transfer learning [2]. For example, Craighero et al. [7] proposed SemiFDA, which trains local feature extractors on clients to align them with the server. In aggregation optimization-based methods, the primary focus is on optimizing aggregation weights [62], gradients [43], and aggregation strategies [41, 6]. For example, FedHEAL [5] removes some less important updates from client models and determines aggregation weights based on the distance between the global model and each local model. AutoFedGP [18] calculates the distance between the source and target domains to derive a new automatic weighting scheme. The aforementioned FDA works are primarily based on the assumption of a closed environment. Currently, there is limited research on FDA in open environments. Even exists, *e.g.*, FOSDA [37], it is only applicable to class-incremental scenarios and does not consider the impact on the source domain.

## 3 Methodology

This section starts with a motivation example and outlines the pipeline of our developed federated domain adaptation scheme, *Gains*. Subsequently, we elaborate on fine-grained knowledge discrimination and contribution-driven knowledge adaptation as two key components in *Gains*.

**Motivation.** Without loss of generality, we use the LeNet model and MNIST dataset as an example, considering a scenario where the first three new clients' data is from the source domain, the fourth introduces 1–4 new classes, and the fifth brings new domain data (details are shown in Appendix. A). When new clients participate in training, the variations of the encoder, classifier, and extracted feature are measured by the distance (*e.g.*, Euclidean distance) before and after training in the target domain. From Fig. 2, we have the following findings: *(i)* the variation of the encoder (*i.e.*, $Diff^E$) does not show a clear fluctuation trend no matter in class or domain incremental scenarios; *(ii)* the changes in the classifier parameters (*i.e.*, $Diff^C$) are more pronounced in the class-incremental scenario; *(iii)* while both new classes and domain will bring obvious changes to the feature values (*i.e.*, $Diff^F$), it is more significant in the domain-incremental scenario. Therefore, it is reasonable to consider a combined evaluation of $Diff^C$ and $Diff^F$ to determine whether the new client introduces new knowledge and whether such knowledge is class- or domain-related.

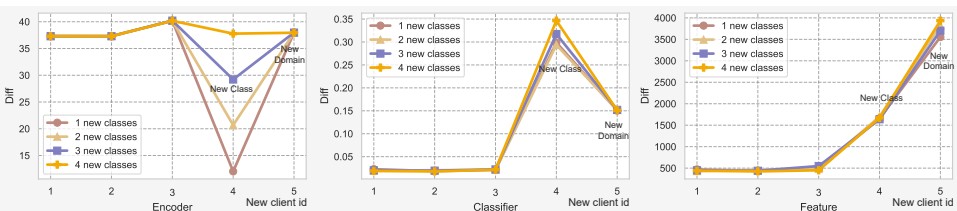

Figure 2: Differences in the encoder (left), classifier (middle), and extracted feature values (right) when the new client carries different types of knowledge.

**Framework.** Inspired by the above empirical discoveries, we propose a fine-grained federated domain adaptation framework in open set, *Gains* (shown in Fig. 3). Specifically, it consists of two main components: knowledge discovery and knowledge adaptation. In the knowledge discovery stage, the target domain performs local training based on the source model and uploads the updated version back to the server. Then, the server uses public dataset to calculate the variations of $Diff^C$ and $Diff^F$, determining whether the new client introduces new knowledge and further discriminating its type in fine grains. Based on the results of this differentiation, in the knowledge adaptation stage, the contribution of different model components in each source model is calculated. After that, the server executes contribution-driven aggregation to accelerate the speed of target domain adaptation. Considering it may lead to an overemphasis on the target domain, potentially resulting in the performance degradation of the source domain, an anti-forgetting mechanism is included in the local training of the source client to balance the knowledge of the target and source domains.

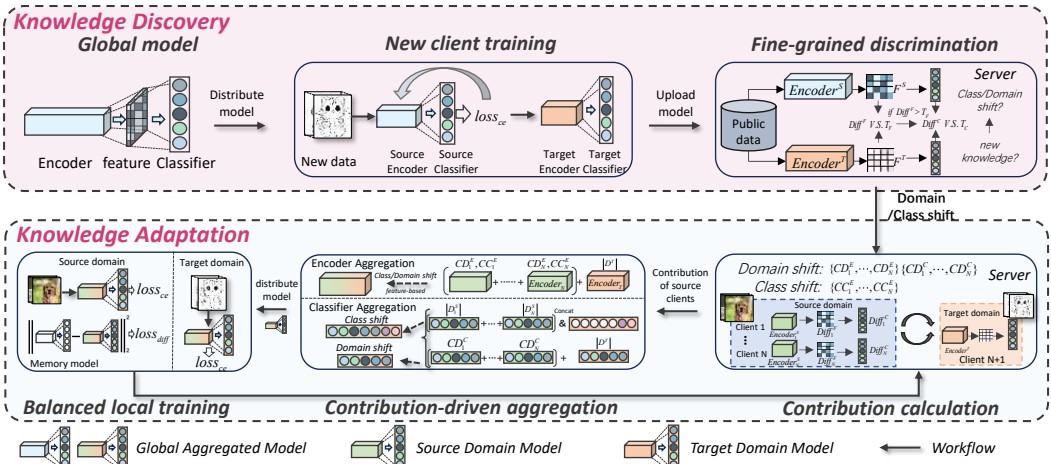

Figure 3: ***Gains*** **consists of two phases: knowledge discovery (upper) and knowledge adaptation (lower).** The former works on identifying the type of new knowledge, while the latter attempts to achieve rapid integration of new knowledge and strike a balance between new and old knowledge.

**Notations.** We denote the $N$ client source domain dataset by $\mathcal{D}_n^S = \left\{ (x_j^n, y_j^n) \big|_{j=0}^{|\mathcal{D}_n^S|} \right\}, n = 1, \cdots N$. The target domain dataset is $\mathcal{D}^T = \left\{ (x_j, y_j) \big|_{j=0}^{|\mathcal{D}^T|} \right\}$. The server's public data is $\mathcal{D}^P = \{ (x^p, y^p) \}$. The original pre-trained source domain model is $\mathcal{W}^S$, comprising an encoder $E^S$ and a classifier $C^S$. Similarly, we write the target domain trained model as $\mathcal{W}^T$, which includes $E^T$ and $C^T$. $I$ is the total federated iteration and $R$ is the local training epoch.

## 3.1 Fine-grained knowledge discrimination

When a new client enters, the server first distributes original source global model $\mathcal{W}^S$ to the target domain for local training $Q$ times. The optimization process is as follows:
$$\mathcal{W}^T(q + 1) = \mathcal{W}^T(q) - \eta \nabla \mathcal{L}(\mathcal{W}^T(q), \mathcal{D}^T), q = 0, \cdots, Q-1, \tag{1}$$
where $\eta$ is learning rate, $\mathcal{L}(\mathcal{W}^T(q), \mathcal{D}^T)$ is the loss in the $q$-th local training epoch, and the final target model is $\mathcal{W}^T = \mathcal{W}^T(\hat{I})$. Once the target client finished the local training, $\mathcal{W}^T$ will be uploaded to the server. Then, $\mathcal{D}^P$ is input into $E^S$ and $E^T$ to obtain the feature values $F^S = E^S(x^p)$ and $F^T = E^T(x^p)$, respectively. According to the finding (iii) in the motivation, we first judge whether the new client brings new knowledge based on the variation of $Diff^F$. If $Diff^F$ is big enough, we believe that the data distribution from the target domain is different from that of the source domain, which means new knowledge is coming. Furthermore, according to finding (ii), we can determine whether this new knowledge is related to a new class by calculating $Diff^C$. Specifically, we use the Manhattan distance and the Euclidean distance to calculate $Diff^F$ and $Diff^C$, respectively.

We set $T_F$ and $T_C$ as thresholds for discovering new knowledge and determining the type of new knowledge, respectively. When $Diff^F > T_F$, we conclude that the target domain has introduced new knowledge. If $Diff^C > T_C$ simultaneously, it indicates that the new knowledge corresponds to a new class; otherwise, it is considered as new domain knowledge.

## 3.2 Contribution-driven knowledge adaptation

In the knowledge adaptation phase, two key issues need to be addressed: first, the rapid knowledge adaptation to the target domain; and second, the balance between new and old knowledge. To achieve the former one, we propose the contribution-driven aggregation strategy, which means assigning greater weights to clients with higher contributions. As for the latter balance problem, an anti-forgetting mechanism is presented.

**Domain-incremental contribution-driven aggregation.** In this paper, we believe that the more similar the source domain client is to the target domain, the more beneficial it is for the fusion of new

knowledge. Then the greater the contribution is. In the domain-incremental scenario, the encoder and the classifier adopt the feature-based and parameter-based contribution calculation methods, respectively. In the feature-based calculation, the encoder contribution $\mathcal{CD}_n^E(i)$ of the $n$-th source client to the target domain during the $i$-th iteration is calculated as follows:

$$\mathcal{CD}_n^E(i) = \frac{1}{(1 + Diff_n^F(i)) \times \sum_{n=1}^{N} \left( 1/(1 + Diff_n^F(i)) \right)} \times \frac{\sum_{n=1}^{N} |\mathcal{D}_n^S|}{|\mathcal{D}^T| + \sum_{n=1}^{N} |\mathcal{D}_n^S|}, \qquad (2)$$

where $Diff_n^F(i)$ is measured by the distance between $F^T(i) = E^T(i)(x^p)$ and $F_n^S(i) = E_n^S(i)(x^p)$. $E^T(i)$ is the encoder uploaded by the target domain in $i$-th iteration, while $E_n^S(i)$ is from the $n$-th source client. Similarly, in parameter-based aggregation, the classifier contribution of $n$-th source client $\mathcal{CD}_n^C(i)$ is calculated as follows:

$$\mathcal{CD}_n^C(i) = \frac{1}{(1 + Diff_n^C(i)) \times \sum_{n=1}^{N} \left( 1/(1 + Diff_n^C(i)) \right)} \times \frac{\sum_{n=1}^{N} |\mathcal{D}_n^S|}{|\mathcal{D}^T| + \sum_{n=1}^{N} |\mathcal{D}_n^S|}, \qquad (3)$$

where $Diff_n^C(i)$ is measured by the distance between $C^T(i)$ and $C_n^S(i)$. Ultimately, we obtain the contribution lists $\{\mathcal{CD}_1^E(i), \mathcal{CD}_2^E(i), \cdots, \mathcal{CD}_N^E(i)\}$ and $\{\mathcal{CD}_1^C(i), \mathcal{CD}_2^C(i), \cdots, \mathcal{CD}_N^C(i)\}$ of the source encoder and source classifier during the $i$-th iteration in the domain-incremental scenario. The aggregation processes are as follows:

$$E(i) = \sum_{n=1}^{N} \mathcal{CD}_n^E(i) \times E_n^S(i) + \frac{|\mathcal{D}^T|}{|\mathcal{D}^T| + \sum_{n=1}^{N} |\mathcal{D}_n^S|} \times E^T(i), \qquad (4)$$

$$C(i) = \sum_{n=1}^{N} \mathcal{CD}_n^C(i) \times C_n^S(i) + \frac{|\mathcal{D}^T|}{|\mathcal{D}^T| + \sum_{n=1}^{N} |\mathcal{D}_n^S|} \times C^T(i). \qquad (5)$$

The aforementioned aggregation process facilitate the rapid adaptation of knowledge by dynamically improving the contribution-based weights in each iteration.

**Class-incremental contribution-driven aggregation.** Similarly, in the class-incremental scenario, for the encoder aggregation process, we adopt the same feature-based method to calculate the contribution. The Encoder contribution of the $n$-th source client in $i$-th iteration is $\mathcal{CC}_n^E(i)$. Then, the contribution list of the encoder in class-incremental scenarios is obtained $\{\mathcal{CC}_1^E(i), \mathcal{CC}_2^E(i), \cdots, \mathcal{CC}_N^E(i)\}$. The aggregation process is as follows:

$$E(i) = \sum_{n=1}^{N} \mathcal{CC}_n^E(i) \times E_n^S(i) + \frac{|\mathcal{D}^T|}{|\mathcal{D}^T| + \sum_{n=1}^{N} |\mathcal{D}_n^S|} \times E^T(i). \qquad (6)$$

The aggregation of the classifier employs a channel-wise supplementation method. First, the classifiers from the source domain are aggregated based on the amount of data from each client, resulting in $C^S(i) = \sum_{n=1}^{\mathcal{N}} \frac{|D_n^S|}{\sum_{n=1}^{N} |D_n^S|} \times C_n^S(i)$. Suppose there are $K^S$ classes in the source domain and $K^T$ new classes added in the target domain. Consequently, the classifier has $K^S + K^T$ channels. The parameters of the classifier aggregated from the source domain are denoted as $C^S(i) = [Channel_1^S, \cdots, Channel_{K^S}^S, Channel_{K^S+1}^S, \cdots, Channel_{K^S+K^T}^S]$, and the parameters of the target domain classifier are denoted as $C^T(i) = [Channel_1^T, \cdots, Channel_{K^s}^T, Channel_{K^s+1}^T, \cdots, Channel_{K^s+K^T}^T]$. In the final aggregated classifier, the channels corresponding to the source domain classes directly adopt the parameters from $C^S(i)$, while the channels for the target domain classes retain the parameters from $C^T(i)$. That is,

$$C(i) = \left[ \underbrace{Channel_1^S, \cdots, Channel_{K^S}^S}_{Source Domain}, \underbrace{Channel_{K^S+1}^T, \cdots, Channel_{K^S+K^T}^T}_{Target Domain} \right]. \qquad (7)$$

A theoretical convergence analysis of *Gains* is provided in the Appendix. F.

**Anti-forgetting mechanism.** The above aggregation may lead to a bias towards the target domain knowledge in the aggregated model, potentially causing a decline in performance on the source domain tasks. To mitigate this, we introduce an anti-forgetting mechanism for the source domain clients during each round of local training. Specifically, we control the distance between the current model $\mathcal{W}_n^S(i, r)$ and the memory model $\mathcal{W}_n^S(0, 0)$ in the local training to prevent the local model

from excessively deviating from the historical model. Here, $\mathcal{W}_n^S(0,0)$ represents the local model in the source domain before the new client enters. $\mathcal{W}_n^S(i,r)$ is the $n$-th client model during the $i$-th global iteration and $r$-th local training epoch. The local loss function for the source clients is defined as follows:

$$\mathcal{L}(\mathcal{W}_n^S(i,r), \mathcal{D}_n^S) = -\frac{1}{|\mathcal{D}_n^S|} \sum_{j=1}^{|\mathcal{D}_n^S|} \sum_{c=1}^{K^S+K^T} y_{j,c}^n \log(\hat{y}_{j,c}^n) + \lambda \left\| \mathcal{W}_n^S(i,r) - \mathcal{W}_n^S(0,0) \right\|_2^2, \tag{8}$$

where $\lambda$ is a balance coefficient. Through the above training process, we can achieve rapid federated domain adaptation while avoiding forgetting the source domain knowledge, thereby maintaining a balance between new and old knowledge.

### 3.3 Algorithm

As shown in Alg. 1, when a new client joins, the server distributes the source global model $\mathcal{W}^S$ to the target domain for local training, getting $\mathcal{W}^T$. Subsequently, the server decomposes $\mathcal{W}^S$ and $\mathcal{W}^T$ into an encoder and a classifier and derives the feature using the public dataset. Based on the differences in the feature extracted by $E^S$ and $E^T$, as well as the parameter differences between $C^S$ and $C^T$, the algorithm discriminates the type of new knowledge and confirms its type. Then, we calculate the contributions of the source clients to the target client in both encoders and classifiers. According to knowledge types and model components, specific aggregation strategies are used to accelerate knowledge adaptation. Furthermore, to prevent the aggregation process from overly favouring the target client, the anti-forgetting mechanism is incorporated into the local update process of the source clients. After all clients complete local training, they upload their models to the server for aggregation based on their contributions. This process repeats until convergence. If no new knowledge is detected at the outset, the original model is deployed directly on the newly joined clients for inference without any further training.

## 4 Experimental verification

This section first explores the threshold for knowledge discovery and validates *Gains* under three data shift scenarios. Then, to verify its scalability, we conduct experiments in more target domains and a sequential FDA scenario. Finally, ablation studies reveal the necessity of the AFM component.

### 4.1 Experiment setting

Our experiments are conducted on a single NVIDIA RTX 4090 GPU. We construct a federated learning framework that includes one server and 50 clients for validation. Following [7], we evaluate *Gains* in three scenarios of target data shifts: mild, medium, and strong shifts. Specifically, under the mild shift scenario, clients in both the source and target domains are drawn from the same sub-dataset but contain different classes. Under the medium shift scenario, all clients in the source domain are from one sub-dataset, while clients in the target domain are from another sub-dataset. Under the strong shift scenario, different clients in the source domain contain different sub-datasets, and clients in the target domain are from other sub-datasets. The main results are shown in Table 1.

**Dataset.** The datasets include the DigitFive (*i.e.*, DF) for the digit classification and the Amazon Review (*i.e.*, AR) for the product review. DF comprises five sub-datasets: MNIST, MNIST-M, SVHN, USPS, and SynthDigits. Each one contains 10 classes of digits from 0 to 9. The AR dataset records user reviews of products on the Amazon website and includes four subdatasets: Books, DVDs, Electronics, and Kitchen housewares. Each sub-dataset contains two classes.

**Baselines.** We include two categories of baselines. The first is to address the domain adaptation problem, including FOSDA [37], SemiFDA [7], AutoFedGP [18] and FedHEAL [5]. The second focuses on the heterogeneous problem, including FedAVG [33], FedProx [26], and FedProto [42].

**Evaluations.** (i) the accuracy of the target client (*T-Acc*); (ii) the average accuracy of the source clients (*S-Acc*); (iii) the global accuracy (*G-Acc*).

### 4.2 New knowledge discovery

The key to discovering new knowledge lies in setting an appropriate threshold, *i.e.*, $T_F$ and $T_C$. In Fig. 2, we observe that when new clients introduce unseen class or domain knowledge, the $Diff^F$

**Algorithm 1:** *Gains*

---

**Input:** Number of source clients $N$; original source global model $\mathcal{W}^S$ and client model $\{\mathcal{W}_1^S(0,0), \mathcal{W}_2^S(0,0), \cdots, \mathcal{W}_N^S(0,0)\}$; number of iteration $I$; number of local training $R$; public data $\mathcal{D}^P = \{(x^p, y^p)\}$

**Output:** Global model $\mathcal{W}$

1 Distribute original source model $\mathcal{W}^S$ to target client

2 $\mathcal{W}^T \leftarrow$ Target client performs local updating based on $\mathcal{W}^S$

3 Target client Uploads $\mathcal{W}^T$ to the server

4 *//Knowledge Discovery*

5 Split the $\mathcal{W}^S$ into encoder $E^S$ and classifier $C^S$, split the $\mathcal{W}^T$ into $E^T$ and $C^T$

6 $F^S \leftarrow E^S(x^p)$, $F^T \leftarrow E^T(x^p)$

7 Calculating $Diff^C$ and $Diff^F$

8 **if** $Diff^F > T_F$ **then**

9       Target client brings new knowledge

10      **if** $Diff^C > T_C$ **then**

11          $Class\ Increment$=True

12      **else**

13          $Domain\ Increment$=True

14      *//Knowledge Adaptation*

15      **for** *iteration* $i = 0, \cdots, I$ **do**

16          **if** $Domain\ Increment$=*True* **then**

17              Calculating encoder contributions $\{\mathcal{CD}_1^E, \mathcal{CD}_2^E, \cdots, \mathcal{CD}_N^E\}$ based on Eq. (2)

18              Calculating classifier contributions $\{\mathcal{CD}_1^C, \mathcal{CD}_2^C, \cdots, \mathcal{CD}_N^C\}$ based on Eq. (3)

19              Aggregating all clients' parameters using Eq.(4) and Eq. (5)

20          **if** $Class\ Increment$=*True* **then**

21              Calculating encoder contributions $\{\mathcal{CC}_1^E, \mathcal{CC}_2^E, \cdots, \mathcal{CC}_N^E\}$ based on Eq. (2)

22              Aggregating all clients' parameters using Eq.(6) and Eq. (7)

23      Server distributes the aggregated model to all clients

24      **for** *client* $n = 1, \cdots, N$ **do**

25          Locally update model $R$ rounds using Eq.(8)

26          Upload $\mathcal{W}_n^S(i, R)$ to the server

27      Target client locally update model $R$ rounds and upload to the server

28 **else**

29      Apply the original model to newly joined clients for inference tasks without training

---

increases significantly, with all values exceeding 1000. Furthermore, in the case of class increment, the $Diff^C$ undergoes substantial changes. Even when only a new class is added to the target client, the parameter change of the classifier is still greater than 0.25, which is significantly higher than that of domain increment clients. Therefore, for the DigitFive dataset, we consider setting the threshold $T_F$ to 1000 and the threshold $T_C$ to 0.25. For the Amazon Review dataset, given the limited number of classes, we only conduct validation in the domain increment scenario. Taking DVDs as the source domain data and Kitchen Hardware as the target domain data as an example, when the new client does not introduce new data, the $Diff^F$ fluctuates between 50 and 150. However, when the new nodes bring in new domain data, the change value increases to 534.76. Therefore, we consider setting the threshold $T_F$ for the Amazon Review dataset to 400.

## 4.3 Knowledge adaptation

**Mild data shift.** Under the mild data shift scenario, we experiment using the MNIST data from the DigitFive dataset, assuming that the target domain contains data labeled as {1, 5}, while the source domain consists of {0, 2, 3, 4, 6, 7, 8, 9}. *Gains* achieves 99.34% new client accuracy (*T-Acc*) while maintaining 93.21% source client accuracy (*S-Acc*) and 94.44% global accuracy (*G-Acc*). This demonstrates *Gains*'s effectiveness in class-incremental scenarios. The feature-based contribution calculation and channel-wise classifier aggregation allow seamless integration of new

Table 1: Main results. The bold font represents the optimal result.

| Scenario | Metric | | Federated Domain Adaptation | | | | Heter-FL | | |
|---|---|---|---|---|---|---|---|---|---|
| | | Ours | FOSDA [TNNLS'24] | SemiFDA [ICDM'24] | AutoFedGP [ICLR'24] | FedHEAL [CVPR'24] | FedAVG [AISTATS'17] | FedProx [MLSys'20] | FedProto [AAAI'22] |
| **DigitFive** | | | | | | | | | |
| **Mild** | T-Acc | **99.34** | 0.00 | 0.00 | 68.11 | 22.60 | 55.73 | 72.35 | 77.61 |
| | S-Acc | 93.21 | 12.72 | 13.53 | 0.00 | 99.29 | 0.36 | **99.53** | 0.16 |
| | G-Acc | **94.44** | 10.18 | 10.83 | 13.62 | 83.95 | 11.44 | 94.09 | 62.12 |
| **Medium** | T-Acc | **97.91** | 11.29 | 7.91 | 9.78 | 93.68 | 90.79 | 94.88 | 45.66 |
| | S-Acc | **90.09** | 19.46 | 19.44 | 6.22 | 88.71 | 76.20 | 86.50 | 33.56 |
| | G-Acc | **91.65** | 17.82 | 17.14 | 6.93 | 89.70 | 79.12 | 88.18 | 43.23 |
| **Strong** | T-Acc | **98.98** | 11.29 | 31.14 | 10.37 | 96.98 | 85.80 | 85.29 | 31.28 |
| | S-Acc | **93.18** | 13.60 | 14.21 | 11.60 | 83.32 | 43.90 | 43.32 | 62.23 |
| | G-Acc | **94.34** | 13.13 | 17.60 | 11.35 | 86.05 | 52.28 | 51.72 | 37.47 |
| **Amazon Review** | | | | | | | | | |
| **Medium** | T-Acc | **84.60** | 49.55 | 50.45 | 50.50 | 50.56 | 66.74 | 74.55 | 50.11 |
| | S-Acc | **82.81** | 49.55 | 49.33 | 50.50 | 50.56 | 67.19 | 74.44 | 50.11 |
| | G-Acc | **83.09** | 49.82 | 49.82 | 50.58 | 50.48 | 67.38 | 74.12 | 50.01 |
| **Strong** | T-Acc | **80.54** | 50.48 | 55.41 | 50.03 | 83.34 | 51.20 | 53.73 | 50.10 |
| | S-Acc | **84.95** | 50.27 | 59.25 | 50.02 | 86.54 | 51.36 | 53.95 | 50.11 |
| | G-Acc | **83.85** | 50.33 | 58.29 | 50.02 | 85.74 | 51.32 | 53.89 | 50.10 |

classes. Meanwhile, the anti-forgetting mechanism further ensures stable source performance by constraining parameter drift during local updates.

**Medium data shift.** For a more complex scenario, medium data shift, we conduct validation using DigitFive and Amazon Review datasets. As for DigitFive, the source domain data is derived from SVHN, while the target domain's data is from MNIST. For Amazon Review, the corresponding data are DVDs and Books, respectively. In Table 1, *Gains* achieves 97.91% *T-Acc* and 90.09% *S-Acc* in DigitFive, outperforming all baselines. Notably, FedHEAL achieves competitive T-Acc (93.68%) but exhibits unstable source performance (*S-Acc*=88.71%). A similar phenomenon can be observed in the Amazon Review dataset. This validates the effect of *Gains* in domain-incremental scenarios: leveraging feature gap in the encoder and parameter variation in the classifier to dynamically prioritize source clients with higher contributions.

**Strong data shift.** In extreme cases, each client in source and target domains may come from different domains, which refer to as strong data shift. For DigitFive, we assume that the target domain client data is from MNIST, and the source domain consists of four clients, each holding MNIST-M, SVHN, USPS, and SynthDigits datasets, respectively. For the Amazon Review, the target domain is Books, and source-domain clients are from the DVDs, Electronics, and Kitchen Housewares datasets. In Table 1, *Gains* achieves 98.98% *T-Acc* and 93.18% *S-Acc* in DigitFive, demonstrating robustness to extreme heterogeneity. Similarly, *Gains* achieves 80.54% *T-Acc*, 84.95% *S-Acc* and 83.85% *G-Acc* in Amonzon Review, showing significant advantages over other methods.

**Adaptation speed.** The above content illustrates that the *Gains* method can improve learning performance in the source and target domains. To further demonstrate its advantage in domain adaptation speed, we visualize the training process of DigitFive under different methods in Fig. 4, where the vertical axis represents the global accuracy and the horizontal axis represents the number of epochs. It can be seen that our method not only achieves the highest accuracy but also has the fastest convergence speed, enabling it to reach better results more quickly. This is because we optimize the aggregation process of the encoder and classifier based on their respective contributions, which allows for more efficient adaptation of new knowledge on the basis of the source domain model. The convergence process diagram for the Amazon Review dataset is provided in the Appendix. C.

**Generalization verification.** In Table 1, we only validate some cases under the mild (Mi), medium (Me), and strong (St) data shift scenarios. To further verify the generalization ability of *Gains*, we change the source/target domain datasets and test the DF and AR datasets under above three scenarios, and the results are shown in Table 2. Here, {1,5} indicates that the target domain data labels are 1 and 5. "SV-MT" represents the scenario where the source domain is SVHN and the target domain is MNIST under the Me data shift. MTM, BK, DD, and KC are the abbreviations for the MNISTM, Book, DVDs, and Kitchen datasets, respectively. As shown in Table 2, under the same multi-domain dataset, our method still maintains a comparable level when the target domain is different, indicating strong generalization capabilities of *Gains*. Please refer to Appendix. D for more validations on the generalization.

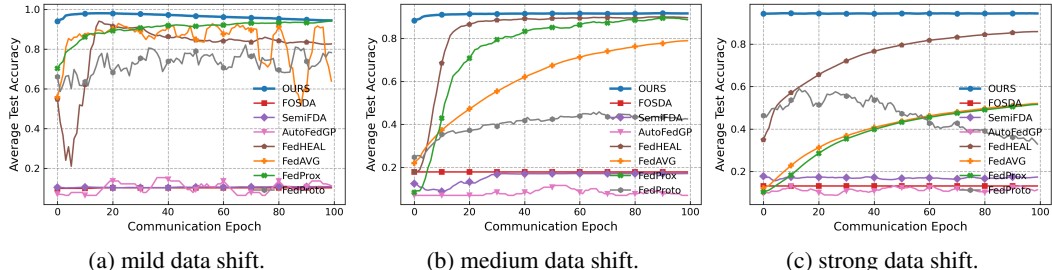

|       |       | (a) mild data shift. |       | (b) medium data shift. |       | (c) strong data shift. |
|-------|-------|----------------------|-------|------------------------|-------|------------------------|

Figure 4: Training process of DigitFive under different data shift scenarios.

**Sequential FDA.** In the previous experiments, we primarily focus on the scenario where only a single new client joins in FL. In this part, we take DigitFive as an example to verify the performance when continuous new clients arrival (*i.e.*, sequential FDA). In the class-incremental scenario, we assume that the source domain classes are {0,1,2,3}, and subsequently, three clients carrying {4,5}, {6,7}, and {8,9} join the FL process. In the domain-incremental scenario, the source domain is the SVHN, and the target domains include MNIST, MNIST-M, and SynthDigits, respectively. Table 3 shows the results after incorporating different target domain data into the training process. It can be observed that *Gains* still exhibits strong robustness in sequential FDA.

**Ablation study.** This part examine the role of the Anti-forgetting Mechanism in *Gains* using DigitFive dataset. As shown in Table 4, the absence of the AFM indeed causes significant performance degradation for the source clients across all scenarios, illuminating the effectiveness of this component. Moreover, the performance drop is most pronounced in the class-incremental scenario (*i.e.*, mild data shift). This is consistent with our observations in the motivation, as the changes to the model parameters are most significant during class increment. Without AFM, in the mild data shift scenario, the client model deviates most severely from its original parameters, resulting in the greatest performance decline.

### 4.4 Computing complexity analysis.

Compared with traditional federated learning, *Gains* mainly increases the computational load during the server-side contribution calculation. Its complexity is $O(N \cdot P \cdot d)$, where $N$ is the number of source domain clients, $P$ is the size of the public dataset, and $d$ is the number of model parameters. Inevitably, extra computational costs occur during the above process. However, by calculating the weights based on contribution, more efficient aggregation can be achieved, thereby significantly reducing the number of federated iterations and reducing the overall training time. Taking the DigitFive dataset in the mild shift scenario as an example, the consumed computing resources and the number of iterations are as shown in Table 5.

### 4.5 Sensitivity analysis of the thresholds

Although the thresholds are manually set, the model exhibits strong robustness to threshold variations. As can be seen from Figure 2, the changes in $Diff^F$ and $Diff^C$ are very significant, which means

Table 2: Generalization verification.

|       |    | {1,5}  | {6,9}   | {0,1,5} |
|-------|----|--------|---------|---------|
|       | TA | 99.34  | 94.42   | 99.59   |
| Mi-DF | SA | 93.21  | 96.03   | 87.16   |
|       | GA | 94.44  | 95.71   | 89.64   |
|       |    | SV-MT  | MT-MTM  | SYN-MTM |
|       | TA | 97.91  | 94.46   | 90.49   |
| Me-DF | SA | 90.09  | 99.56   | 98.57   |
|       | GA | 91.65  | 98.54   | 96.95   |
|       |    | MT     | SV      | MTM     |
|       | TA | 98.98  | 91.67   | 93.94   |
| St-DF | SA | 93.18  | 97.58   | 96.20   |
|       | GA | 94.34  | 96.40   | 95.75   |
|       |    | DD-BK  | BK-DD   | ET-KC   |
|       | TA | 84.60  | 82.01   | 86.59   |
| Me-AR | SA | 82.81  | 86.85   | 89.93   |
|       | GA | 83.09  | 85.88   | 89.26   |
|       |    | BK     | DD      | KC      |
|       | TA | 80.54  | 78.22   | 85.38   |
| St-AR | SA | 84.95  | 88.90   | 87.73   |
|       | GA | 83.85  | 86.23   | 87.14   |

Table 3: The performance of sequential FDA.

|    |    | {4,5}  | {6,7}   | {8,9}   |
|----|----|--------|---------|---------|
|    | TA | 99.88  | 91.35   | 96.89   |
| Mi | SA | 93.53  | 99.43   | 99.35   |
|    | GA | 96.82  | 98.08   | 99.00   |
|    |    | MNIST  | MNISTM  | SYN     |
|    | TA | 95.27  | 83.53   | 93.53   |
| Me | SA | 87.91  | 90.05   | 89.66   |
|    | GA | 89.38  | 88.96   | 90.21   |

Table 4: Ablation study of AFM.

|         | Mild  | Medium | Strong |
|---------|-------|--------|--------|
| AFM     | 99.05 | 90.09  | 94.77  |
| w/o AFM | 9.24  | 84.35  | 92.46  |

that the thresholds can take values over a wide range, with $Diff^F$ ranging from 700 to 3400 and $Diff^C$ from 0.05 to 0.27. We also conduct experiment tests on various thresholds using the mild shift scenario in the DigitFive dataset as an example. Assuming the source domain data is MNIST-M and the target domain data is MNIST, with $T_F \in 800, 1000, 1200$ and $T_C \in 0.20, 0.25, 0.27$. The experimental results obtained are shown in Table and Table...From the above two tables, it can be seen that the model performance remains stable when parameters fluctuate within reasonable ranges (performance variation < 1%).

## 5 Conclusion and discussion

**Conclusion.** This paper presents a novel fine-grained federated domain adaptation framework in open set (*Gains*) that addresses the challenges of fine-grained knowledge discovery and rapid and balanced knowledge adaptation. By splitting the model into an encoder and a classifier, *Gains* effectively identifies the type new knowledge based on the variations in extracted features and model parameters, enabling more precise knowledge adaptation. The proposed contribution-driven aggregation strategy accelerates the integration of new knowledge into the global model, while the anti-forgetting mechanism ensures the preservation of source domain performance. Extensive experiments on multiple datasets demonstrate that *Gains* can achieve balanced adaptation and rapid convergence under various data shift scenarios.

**Discussion.** This paper proposes a fine-grained domain adaptation framework in FL. Although the pipeline achieves satisfactory results, some limitations still exist. First, in the knowledge discovery phase, it still relies on manually set thresholds, and achieving automatic knowledge discovery remains a significant challenge. Second, in the knowledge identification phase, we consider domain increment and class increment. However, for more complex scenarios, such as task increment or scenarios

Table 5: Convergence Comparison of Different Methods.

| Method | Converge Round | Time |
|---|---|---|
| Gains | **5** | **807.45** |
| FedHEAL | 40 | 1368.4 |
| FedAVG | 20 | 1977.20 |
| FedProx | 40 | 6880.80 |
| FedProto | 32 | 9519.68 |

Table 6: Accuracy Results for Different $T_F$ Values.

| $T_F$ | T-Acc | S-Acc | G-Acc |
|---|---|---|---|
| 800 | 99.62 | 92.01 | 93.15 |
| 1000 | 99.34 | 93.21 | 94.44 |
| 1200 | 99.24 | 93.06 | 93.91 |

Table 7: Accuracy values for different $T_C$ settings.

| $T_C$ | T-Acc | S-Acc | G-Acc |
|---|---|---|---|
| 0.20 | 99.75 | 92.34 | 93.29 |
| 0.25 | 99.34 | 93.21 | 94.44 |
| 0.27 | 99.86 | 92.71 | 93.01 |

involving both class increment and domain increment, further exploration is needed. In addition, it's worth noting *Gains* is significantly different from traditional federated continual learning. First, the settings are different. FCL primarily focuses on scenarios where existing clients encounter new data, while FDA focuses on cases where new clients join and bring unseen data. Second, the objectives are different. FCL primarily addresses the catastrophic forgetting caused by new data in existing clients. In contrast, *Gains* focuses on rapidly adapting to the new domain while preventing performance degradation of the source domain clients, achieving efficient and balanced domain adaptation.

**Acknowledgment.** This work is supported by National Natural Science Foundation of China under Grant 62273352 and 72501290.

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

## A   Motivation experiment settings

In the motivation experiments, we treat the last layer of LeNet as the classifier and the remaining layers as the encoder. In the class-incremental scenario, the source domain data contains classes {3, 4, 6, 7, 8, 9}. After source-domain training is completed, we sequentially introduce new clients, where the first three are from the source domain, the fourth contains new classes, and the fifth is from a new domain. Under the class-incremental setting, we consider the target client data classes to be {5}, {1, 5}, {0, 1, 5}, and {0, 1, 2, 5}, corresponding to the addition of 1, 2, 3, and 4 classes, respectively. In the domain-incremental scenario, the target domain is the SVHN dataset.

## B   Experimental details

During the experiment, the model used for the DigitFive [1] dataset is a CNN model, while the model used for the Amazon Review dataset [2] is an LSTM. The corresponding hyperparameters for the two datasets are as follows:

Table 8: Hyperparameter setting.

|  | Learning Rate | Optimizer | Batch Size |
|---|---|---|---|
| DigitFive | 0.005 | SGD | 128 |
| Amazon Review | 0.5 | SGD | 64 |

The public dataset used by the server for new knowledge discovery is collected from open sources and typically includes various types of globally known data. Under the scenarios of mild data shift and medium data shift, after determining the data classes contained in the source domain clients, we split the data using the Dirichlet distribution with a hyperparameter of 0.1.

## C   Adaptation speed of Amazon Review

Fig. 5 presents the training process of *Gains* and other baselines on the Amazon Review dataset under the medium data shift and strong data shift scenarios. As shown in the figure, *Gains* achieves convergence in global performance with only a small number of epochs. This further indicates that *Gains* can accelerate the target domain adaptation process and more rapidly integrate target domain knowledge into the global model.

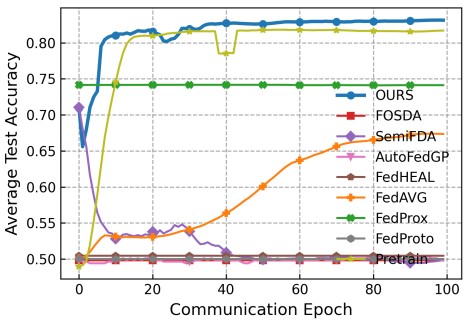 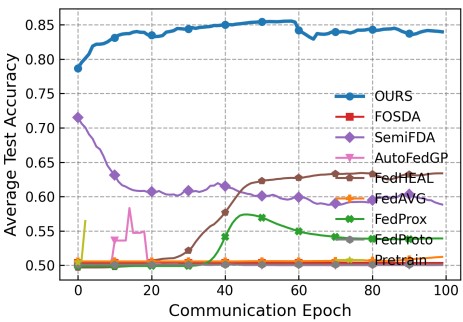

(a) Amazon Review, medium data shift.      (b) Amazon Review, strong data shift.

Figure 5: Training process of Amazon Review under different data shift scenarios.

---

[1]https://ai.bu.edu/M3SDA

[2]https://nijianmo.github.io/amazon/index.html

# D More validations on generalization

In the main manuscript, we validated the effectiveness of *Gains* on part of the cases under three data shift scenarios. In this part, we will verify all the cases under medium data shift and strong data shift, further supporting the generalization capability of *Gains*. Table 9 and Table 11 show the results under different data shift scenarios for the DigitFive dataset, while Table 10 and Table 12 present the results for the Amazon Review dataset under similar conditions. It can be observed that for the DigitFive dataset, the *T-Acc*, *S-Acc*, and *G-Acc* all exceed 90% across all scenarios. Similarly, for the Amazon Review dataset, the *T-Acc*, *S-Acc*, and *G-Acc* are mostly above 80%.

Table 9: DigitFive, medium data shift.

| Source domain | Target domain | T-Acc | S-Acc | G-Acc |
|---|---|---|---|---|
| MNIST-M | USPS | 93.39 | 96.48 | 95.86 |
| MNIST | USPS | 92.10 | 99.50 | 98.02 |
| SynthDigits | USPS | 98.55 | 98.30 | 98.35 |
| SVHN | USPS | 90.16 | 90.78 | 90.65 |
| USPS | MNIST-M | 93.42 | 99.25 | 98.08 |
| MNIST | MNIST-M | 94.46 | 99.56 | 98.54 |
| SynthDigits | MNIST-M | 90.49 | 98.57 | 96.95 |
| SVHN | MNIST-M | 90.97 | 92.38 | 91.30 |
| USPS | MNIST | 98.99 | 99.41 | 99.33 |
| MNIST-M | MNIST | 99.07 | 97.89 | 98.12 |
| SynthDigits | MNIST | 98.89 | 98.59 | 98.65 |
| SVHN | MNIST | 97.91 | 90.09 | 91.65 |
| USPS | SynthDigits | 94.16 | 99.30 | 98.27 |
| MNIST-M | SynthDigits | 92.76 | 97.38 | 96.45 |
| MNIST | SynthDigits | 92.23 | 99.53 | 98.07 |
| SVHN | SynthDigits | 92.69 | 91.10 | 91.42 |
| USPS | SVHN | 92.19 | 99.20 | 95.79 |
| MNIST-M | SVHN | 92.52 | 96.94 | 94.06 |
| MNIST | SVHN | 93.25 | 99.30 | 95.68 |
| SynthDigits | SVHN | 91.69 | 98.79 | 97.37 |

Table 10: Amazon Review, medium data shift.

| Source domain | Target domain | T-Acc | S-Acc | G-Acc |
|---|---|---|---|---|
| Books | Kitchen | 82.22 | 86.43 | 85.59 |
| DVDs | Kitchen | 83.16 | 86.36 | 85.72 |
| Electronics | Kitchen | 86.59 | 89.93 | 89.26 |
| Kitchen | Books | 77.54 | 88.97 | 86.68 |
| DVDs | Books | 80.13 | 83.83 | 83.09 |
| Electronics | Books | 76.37 | 88.36 | 85.97 |
| Kitchen | DVDs | 77.36 | 89.94 | 87.42 |
| Books | DVDs | 82.01 | 86.85 | 85.88 |
| Electronics | DVDs | 77.50 | 88.65 | 86.42 |
| Kitchen | Electronics | 85.55 | 89.67 | 88.85 |
| Books | Electronics | 77.66 | 87.95 | 85.89 |
| DVDs | Electronics | 82.75 | 87.40 | 86.47 |

Table 11: DigitFive, strong data shift.

| Source domain | Target domain | T-Acc | S-Acc | G-Acc |
|---|---|---|---|---|
| MNIST-M, MNIST, SynthDigits, SVHN | USPS | 98.49 | 95.56 | 96.14 |
| USPS, MNIST, SynthDigits, SVHN | MNIST-M | 93.94 | 96.20 | 95.75 |
| USPS, MNIST-M, SynthDigits, SVHN | MNIST | 98.98 | 93.18 | 94.34 |
| USPS, MNIST-M, MNIST, SVHN | SynthDigits | 97.02 | 95.96 | 96.17 |
| USPS, MNIST-M, MNIST, SynthDigits | SVHN | 91.67 | 97.58 | 96.40 |

Table 12: Amazon Review, strong data shift.

| Source domain | Target domain | T-Acc | S-Acc | G-Acc |
|---|---|---|---|---|
| Books, DVDs, Electronics | Kitchen | 85.38 | 87.73 | 87.14 |
| Kitchen, DVDs, Electronics | Books | 80.54 | 84.95 | 83.85 |
| Kitchen, Books, Electronics | DVDs | 78.22 | 88.90 | 86.23 |
| Kitchen, Books, DVDs | Electronics | 86.32 | 85.61 | 85.79 |

## E   Broader impact

This paper is the first to propose a fine-grained knowledge discovery and integration pipeline in the FDA. It can significantly enhance the autonomous evolution capabilities of distributed nodes in open environments without human intervention. Additionally, we have open-sourced our code for reference in future work.

## F   Theoretical analysis

In this subsection, we will analyze the convergence of *Gains* using domain-increment as an example. The following assumptions are made:

**Assumption 1** (Smoothness and Strong Convexity). *The local loss function convex and M-smooth. Then, we have:*

- **M-smoothness:** $\forall \mathcal{W}_n^S(i, e+1), \mathcal{W}_n^S(i, e),$

$$\mathcal{L}(\mathcal{W}_n^S(i, r+1)) - \mathcal{L}(\mathcal{W}_n^S(i, r)) - \left\langle \nabla \mathcal{L}(\mathcal{W}_n^S(i, r)), \mathcal{W}_n^S(i, r+1) - \mathcal{W}_n^S(i, r) \right\rangle$$
$$\leq \frac{M}{2} \left\| \mathcal{W}_n^S(i, r) - \mathcal{W}_n^S(i, r+1) \right\|_2^2$$

.

**Assumption 2** (Smoothness and Strong Convexity). *As the number of iterations increases, the contributions of each source domain client to the target domain gradually become stable.*

The other assumptions are the same as those in Reference [28]. We first analyze the convergence of the Encoder. During each round of global update, the global parameters are:

$$\mathcal{W}(i+1) = \mathcal{W}(i) - \eta \left( \sum_{n=1}^{\mathcal{N}} \mathcal{CD}_n^E(i) \nabla \mathcal{L}\left(\mathcal{W}_n^S(i, R)\right) + \beta(i) \nabla \mathcal{L}\left(\mathcal{W}^T(i, R)\right) \right).$$

Under the smoothness assumption, if the local loss functions of the clients are convex and M-smooth, then the global loss function is also convex and M-smooth, yielding the following result:

$$\mathcal{L}(\mathcal{W}(i+1)) - \mathcal{L}(\mathcal{W}(i)) - \left\langle \nabla \mathcal{L}(\mathcal{W}(i)), \mathcal{W}(i+1) - \mathcal{W}(i) \right\rangle \leq \frac{M}{2} \left\| \mathcal{W}(i) - \mathcal{W}(i+1) \right\|_2^2.$$

Let $\mathcal{W}(i) = -\eta \left( \sum_{n=1}^{\mathcal{N}} \mathcal{CD}_n^E(i) \nabla \mathcal{L}\left(\mathcal{W}_n^S(i, R)\right) + \beta(i) \nabla \mathcal{L}\left(\mathcal{W}^T(i, R)\right) \right) = -\eta \nabla \mathcal{L}(\mathcal{W}(i))$ where

$\beta(i) = \frac{|\mathcal{D}^T|}{|\mathcal{D}^T| + \sum_{n=1}^{\mathcal{N}} |\mathcal{D}_n^S|}$, we can get:

$$\mathcal{L}(\mathcal{W}(i+1)) - \mathcal{L}(\mathcal{W}(i)) + \eta \left\langle \nabla \mathcal{L}(\mathcal{W}(i)), \nabla \mathcal{L}(\mathcal{W}(i)) \right\rangle \leq \frac{M\eta^2}{2} \left\| \nabla \mathcal{L}(\mathcal{W}(i)) \right\|_2^2.$$

For simplicity,

$$\mathcal{L}(\mathcal{W}(i+1)) - \mathcal{L}(\mathcal{W}(i)) \leq \frac{M\eta^2}{2} \left\| \nabla \mathcal{L}(\mathcal{W}(i)) \right\|_2^2 - \eta \left\| \nabla \mathcal{L}(\mathcal{W}(i)) \right\|_2^2.$$

From the above equation, it can be derived that to ensure the total loss value decreases with each iteration, $\eta - \frac{M\eta^2}{2} > 0$ must be satisfied. Therefore, after $I$ times of iterations, we get:

$$\sum_{i=0}^{I} \left\| \nabla \mathcal{L}(\mathcal{W}(i)) \right\|_2^2 \leq \frac{\mathcal{L}(\mathcal{W}(0)) - \mathcal{L}(\mathcal{W}(I))}{\left( \frac{M\eta^2}{2} - \eta \right)}.$$

Since $\mathcal{L}(\mathcal{W}(I)) > 0$, the following conclusion is obtained:

$$\frac{1}{I}\sum_{i=0}^{I}\|\nabla\mathcal{L}(\mathcal{W}(i))\|_2^2 \le \frac{2\mathcal{L}(\mathcal{W}(0))}{I(M\eta^2 - 2\eta)}.$$

Then when $I \to \infty$,

$$\lim_{I\to\infty}\frac{1}{I}\sum_{i=0}^{I}\|\nabla\mathcal{L}(\mathcal{W}(i))\|_2^2 = 0.$$

This indicates that as the number of iterations increases, the global gradient norm tends towards zero, thereby ensuring the convergence of the algorithm. The convergence analysis of the Classifier is similar to that of the Encoder and will not be reiterated here.

