# OpenReview forum: "Gains: Fine-grained Federated Domain Adaptation in Open Set"
_NeurIPS.cc/2025/Conference — NeurIPS 2025 poster_

### Official Review · Reviewer_AsDt · 2025-06-30

**Clarity:** 2
**Significance:** 3
**Originality:** 2
**Rating:** 4
**Confidence:** 4

**Summary:**

The paper proposes Gains, a framework for fine-grained federated domain adaptation in open-set settings. It detects and distinguishes between domain and class shifts by analyzing changes in feature representations and classifier parameters. Gains then adapts the global model using contribution-based aggregation while preserving source performance with an anti-forgetting mechanism. Experiments on benchmark datasets are conducted to show the effectiveness of the proposed method under various data shift scenarios.

**Questions:**

1. The proposed fine-grained knowledge discovery relies on manually set thresholds (TF and TC) for DiffF and DiffC. How sensitive is the method to these thresholds? Could the authors propose a principled or automated way to set them, possibly based on validation metrics or adaptive criteria?
2. The server uses a public dataset to measure feature and classifier differences. In many federated settings, such datasets may not be available or may violate privacy assumptions. Can the authors clarify what constitutes the “public dataset,” and how the method would perform or adapt without it?
3. While the empirical trends for DiffF and DiffC are shown, there is no theoretical reasoning for why these metrics reliably distinguish class vs. domain increments. Can the authors provide a formal justification or deeper empirical analysis (e.g., statistical separability) to support this design choice?
4. Since the problem setting overlaps with FCL (e.g., handling class increments), why weren’t established FCL baselines like FedWeIT, FedCIL, or FedFCL included in the comparison?
5. The evaluations are based on small models (CNN, LSTM). How does Gains scale to modern architectures like ResNet, ViT, or BERT, especially in terms of DiffF/DiffC computation and aggregation overhead?

**Ethical Concerns:**

["NO or VERY MINOR ethics concerns only"]

**Final Justification:**

The authors addressed my concerns. Therefore, I raise my score.

**Limitations:**

The authors acknowledge key limitations, such as reliance on manually set thresholds and the focus on only domain/class increments, omitting more complex scenarios (e.g., simultaneous or task increments).

**Paper Formatting Concerns:**

Fine

**Quality:**

2

**Strengths And Weaknesses:**

Strengths:
1. The paper addresses federated domain adaptation (FDA) in open-set settings with fine-grained knowledge discovery (distinguishing class vs. domain increments), which is underexplored.

Weaknesses:
1. Lack of theoretical justification for the discrimination mechanism: While convergence of the adaptation strategy is analyzed, there is no theoretical backing for why DiffF and DiffC reliably differentiate domain vs. class shifts. The method might fail in high-dimensional or complex domains (e.g., vision transformers, NLP models) where parameter variation is harder to interpret.
2. Unfair comparisons:  FOSDA and SemiFDA are not inherently designed for class+domain incremental scenarios, so benchmarking them directly without adaptation may not be valid. No strong baselines from federated continual learning (FCL) or class-incremental learning (CIL) are included (e.g., FedWeIT, FedCIL, FedFCL, etc.).

---

> ### Author Rebuttal · Authors · 2025-07-30
>
> We'd like to express our sincere gratitude to the reviewer for your careful reading and highly valuable feedback, which has been extremely beneficial to us. We have made every effort to address your concerns, and some of them will be included in the final version. The following is the the details:
>
> ## Q1. Analysis of the $Diff$ thresholds
> ### (1) Sensitivity analysis
> Although the thresholds are manually set, the model exhibits strong robustness to threshold variations. As shown in Fig. 2, the changes in $Diff^F$ and $Diff^C$ are very significant, which means that the thresholds can take values over a wide range, with $Diff^F$ ranging from 700 to 3400 and $Diff^C$ from 0.05 to 0.27. We also conduct experiments on various thresholds under the mild shift scenario of DigitFive (source: MNIST-M; target: MNIST) as an example, with $T_F \in {800, 1000, 1200}$ and $T_C \in {0.20, 0.25, 0.27}$. The results are as follows:
>
> |$T_F$|T-Acc|S-Acc|G-Acc|
> |-|-|-|-|
> |800|99.62|92.01|93.15|
> |1000|99.34|93.21|94.44|
> |1200|99.24|93.06| 93.91|
>
> |$T_C$|T-Acc|S-Acc|G-Acc|
> |-|-|-|-|
> |0.20|99.75|92.34|93.29|
> |0.25|99.34|93.21|94.44|
> |0.27|99.86|92.71|93.01|
>
> It can be seen that the model performance remains stable when parameters fluctuate within reasonable ranges (performance variation < 1%).
>
> ### (2) Alternative solutions
> To automate the selection of thresholds, we can use a statistics-based approach. In the pre-experiment, we repeatedly introduce clients that carry new knowledge into the federated-learning system and, each time, record the resulting $Diff^F$ and $Diff^C$ values. After collecting many such samples, we compute the means ($\mu_F$, $\mu_C$) and standard deviations ($\sigma_F$, $\sigma_C$) of the two metrics. We then set the thresholds as:
> - $T_F = \mu_F + \alpha\sigma_F$,
> - $T_C = \mu_C + \beta\sigma_C$,
>
> where $\alpha$ and $\beta$ are tunable coefficients that control how strict the thresholds should be.
>
> ## Q2. Concerns about the public dataset
> Public datasets are commonly utilized in FL to simulate shared server knowledge and enable cross-domain evaluations—e.g., FCCL[1] and SacFL[2] both rely on such resources to facilitate client model alignment. Our use aligns with this widespread and pragmatic practice.
>
> In our experiments, the public data is sampled from the DigitFive and Amazon Review datasets, which are strictly disjoint from client data and only stored on the server. Generally speaking, it can be accessed from other datasets that are totally different from clients' data, such as Office-Home, thereby avoiding privacy leakage.
>
> To address scenarios without public datasets, a lightweight client-encoder distillation mechanism can be adopted. Specifically, the server can collect statistical summaries (e.g., encoder outputs’ mean and covariance) from clients instead of raw data, ensuring privacy. By comparing these anonymized feature statistics, the server can approximate $Diff^F$ and $Diff^C$ for knowledge discovery. This part will be shown in the experimental setting and discussion.
>
> - [1] Huang W, et al. Learn from others and be yourself in heterogeneous federated learning[C]//CVPR. 2022: 10143-10153.
>
> - [2] Zhong Z, et al. SacFL: Self-Adaptive Federated Continual Learning for Resource-Constrained End Devices[J]. TNNLS, 2025.
>
> ## Q3 & W1. Concerns about new knowledge discovery mechanism
> ### (1) Empirical analysis
> We conducted a detailed analysis of the DigitFive dataset. In the class increment scenario, the source domain data is MNIST with classes {0,1,2,3}, and we sequentially add data from {0}, {1}, {2}, {3}, {4}, {5}, {6}, {7}, {8}, {9}, where the first four cases correspond to no increment and the latter six cases correspond to class increment. In the domain increment scenario, the source domain is SVHN, and we sequentially add MNIST, MNIST-M, and SynthDigits data. We obtained the mean and variance of $Diff^F$ and $Diff^C$ under different conditions, as shown in the table below:
>
> |Type|Avg. $Diff^F$|Avg. $Diff^C$|
> |-|-|-|
> |No Increment|143.28|0.08|
> |Domain Increment|2897.64|0.11|
> |Class Increment|1832.53|0.29|
>
> As shown above, $Diff^F$ increases dramatically during domain and class increments, while $Diff^C$ exhibits clear peaks during class increments. Using a simple decision boundary (e.g., $Diff^F$ > 1000 → new knowledge, $Diff^C$ > 0.25 → new class), we can easily distinguish these two knowledge increment scenarios.
> ### (2) Theoretical supports
> This work is based on experimental findings that the Decoder (especially the classification head) is more sensitive to changes in classes, while the features extracted by the Encoder are sensitive to changes in knowledge. Therefore, we detect new knowledge by observing changes in the features extracted by the Encoder and the parameters of the Decoder. This approach is well supported by previous studies. For example, Rebuffi et al. [1] found that an increase in the number of classes leads to changes in the structure and parameters of the classifier, indicating that the classifier is sensitive to class changes. Zhong et al. [2] proposed that when new knowledge is introduced, the features extracted by the Encoder undergo significant changes. Therefore, the findings of previous research provide strong theoretical support for the knowledge discovery method proposed in this paper.
>
> - [1] Rebuffi S A, Kolesnikov A, Sperl G, et al. icarl: Incremental classifier and representation learning[C]//CVPR. 2017: 2001-2010.
>
> - [2] Zhong Z, Bao W, Wang J, et al. SacFL: Self-Adaptive Federated Continual Learning for Resource-Constrained End Devices[J]. TNNLS, 2025.
> ## Q4 & W2. Concerns about baselines
> ### (1) FOSDA and SemiFDA
> Although FOSDA and SemiFDA are not specifically designed for class+domain shift scenarios, in our experiments, our method has been compared with them in both individual class increment and domain increment scenarios (Table 1), demonstrating excellent superiority. This further illustrates that Gains possesses good adaptation capabilities.
> ### (2) FCL baselines
> While our work is primarily situated in the open-set federated domain adaptation (FDA) setting, which differs in several key aspects from FCL (illustrated in the discussion), we agree that including FCL baselines addressing class-incremental scenarios would enrich our evaluation. To address this, we conducted additional experiments incorporating FedWeIT and FedCIL as baselines in the class-incremental scenario under mild data shift (DigitFive dataset, target client introduces new classes {1,5}).
>
> |Method|T-Acc|S-Acc|G-Acc|
> |-|-|-|-|
> |Gains (Ours)|99.34|93.21|94.44|
> |FedWeIT|76.22|86.03|84.14|
> |FedCIL|69.85|88.14|83.39|
>
> Gains achieves significantly higher T-Acc while maintaining S-Acc, demonstrating both effective new class integration and source knowledge retention. These baselines will be included in the final version.
>
> ## Q5 & W1. More complex models
> We test the ResNet18 model in the DigitFive dataset under the medium data shift scenario (source: SVHN, target: MNIST). All baselines are reimplemented with ResNet18 as backbone for fair comparison.
>
> |Method|T-Acc|S-Acc|G-Acc|Convergence time|
> |-|-|-|-|-|
> |Gains (Ours)|98.50|93.47|94.47|3528.98|
> |FOSDA|11.29|19.46|17.82|-|
> |SemiFDA|12.31|9.76|10.27|-|
> |AutoFedGP|9.78|6.22|6.93|-|
> |FedHEAL|97.52|92.21|93.27|3113|
> |FedAVG|56.92|69.20|65.23|12568|
> |FedProx|52.63|71.41|66.52|20101.80|
> |FedProto|9.76|6.93|7.50|14890|
>
> We can see that Gains maintains strong generalization on both source and target domains. It outperforms all baselines across all metrics, confirming robustness across architecture types. Meanwhile, under the condition of equivalent learning effectiveness, the computational resources consumed by Gains are lower than those of most other methods (comparable to FedHEAL). Although the calculation of server-side contribution degree will consume more computing resources during the aggregation compared to traditional FL, this process will accelerate the model convergence, thereby reducing the training time. Therefore, it will not result in significant computational resource overhead. This part will be included in the final version.
>
> ## L1. Experiments in complex scenarios
> When both domain shift and class shift occur simultaneously in the target domain, our method still works. We can combine Domain-incremental and Class-incremental contribution-driven aggregation strategies. The aggregation strategy is as follows:
> - Encoder: Contribution calculated by Eq. (2), aggregated by Eq. (4).
> - Classifier:
>   - Old-class channels aggregated via Eq. (3) and Eq. (5),
>   - New-class channels are directly adopted from the target domain model, as described in Line 176 of the manuscript.
>
> To this end, we validated the scenario where both class increment and domain increment occur simultaneously. Specifically, using the DigitFive dataset medium shift as an example, the source domain client data is MNIST with classes {0,2,3,4,6,7,8,9}; the target client includes two types of data: one is MNIST classes {1,5}, and the other is SVHN dataset classes {0,2,3,4,6,7,8,9}. The experimental results obtained are as follows:
>
> |Method|T-Acc|S-Acc|G-Acc|
> |-|-|-|-|
> |Gains-domain+class shift|97.53|91.88|92.91|
> |Gains-domain shift|97.91|90.09|91.65|
> |Gains-class shift|99.34|93.21|94.44|
>
> The results indicate that under the mixed incremental scenario, the performance achieved by Gains is comparable to scenarios with only domain or class increments, without significant degradation. For more complex task-increment scenarios, such as extending from image classification tasks to object detection tasks, adaptive architectures can be designed by incorporating meta-learning techniques. This allows the model to autonomously identify and handle task space expansion, automatically differentiate parameters, and adapt independently, thereby ensuring continual knowledge growth and effective control of unlearning. We will include the above experimental validation and discussion in the final version.

---

> > ### Comment · Reviewer_AsDt · 2025-08-06
> >
> > Thank you to the authors for the rebuttal. While they have addressed the majority of my points, their answer to Question 3 still does not fully convince me.

---

> ### Author Response · Authors · 2025-08-06
> **Discussions about Q3**
>
> Thank you sincerely for your valuable comments. We are sorry for not providing a clear clarification. In the previous version, we had already conducted a thorough empirical analysis. Hence, we made the best efforts to conduct a theoretical analysis in this version and kindly hope this justification addresses the issue you raised. Your discussion is greatly appreciated.
>
> ## **1. Justification Objective**
>
> We aim to prove:
>
> **Discovery 1**. Both domain increment and class increment significantly disturb the encoder $E$'s feature extraction, causing $Diff^F$ increases.
>
> **Discovery 2**. In contrast, class increment leads to a greater disturbance in classifier parameters $C$, i.e., $Diff^C_{class}>Diff^C_{domain}$.
>
> ## **2. Analysis of Discovery 1 ($Diff^F$)**
>
> ### **2.1 Basic Definitions and Assumptions**
> **Definitions**
>
> Let:
>
> - $x \in D_P$: Public input sample from server
> - $E^S$, $E^T$: Encoders from source and target domains
> - Feature change is defined as:
>
>     $$Diff^F = \frac{1}{||D_P||} \sum_{x \in D_P} ||E^T(x) - E^S(x)||_1.$$
>
> **Assumptions**
>
> - The Original input distribution is $\mathbb{P}_S(x)$, and the new client input distribution is $\mathbb{P}_T(x)$, where:
>
>     $\text{domain shift: } \mathbb{P}_T(x) \neq \mathbb{P}_S(x)$ or   $\text{class shift: } \mathbb{P}_T(y) \neq \mathbb{P}_S(y)$.
>
>
> - Encoder is $E$ is $L_E$-Lipschitz.
>
> We aim to analyze why this quantity increases in both class and domain increment scenarios.
>
> ### **2.2 Domain/Class increment**
> #### **Domain increment case**
>
> Since $E$ is $L_E$-Lipschitz, we can get:
>
> $$||E(x_1) - E(x_2)||_1 \leq L_E \cdot ||x_1 - x_2||_1.$$
>
> Using the Wasserstein-1 distance:
>
> $$
> W_1(\mathbb{P}\_T, \mathbb{P}\_S) := \inf_{\gamma \in \Pi(\mathbb{P}\_T, \mathbb{P}\_S)} \mathbb{E}\_{(x,x') \sim \gamma} ||x - x'||_1.
> $$
>
> It measures the distance between two domains. Then, from Kantorovich-Rubinstein Duality, for Lipschitz $f$:
>
> $$
> ||\mathbb{E}\_{x \sim \mathbb{P}\_T}[f(x)] - \mathbb{E}\_{x \sim \mathbb{P}\_S}[f(x)]|| \leq L\_f \cdot W_1(\mathbb{P}\_T, \mathbb{P}\_S).
> $$
>
> Let $f(x) = ||E^T(x) - E^S(x)||_1$, thus:
> $$
> \mathbb{E}{x \sim D_P}[||E^T(x) - E^S(x)||_1] \propto L_E \cdot W_1(\mathbb{P}_T, \mathbb{P}_S).
> $$
>
> As long as $\mathbb{P}_T \ne \mathbb{P}_S$ (i.e., domain increment occurs), the Wasserstein distance $W_1 > 0$, which leads to a significant increase in the feature discrepancy, i.e., $Diff^F$ rises sharply.
>
> #### **Class increment case**
>
> Even if input style remains unchanged $( \mathbb{P}\_T(x) = \mathbb{P}\_S(x) )$, new labels emerge:
>
> - Original labels: $\mathcal{Y}\_S = \{1,...,K\}$
> - New client labels: $\mathcal{Y}\_T = \mathcal{Y}\_S \cup \Delta \mathcal{Y}$
>
> Training on new labels:
> $$
> \min_{E, C} \mathbb{E}\_{(x,y) \sim \mathbb{P}\_T(x,y)} \left[ \mathcal{L}\_{\text{CE}}(C(E(x)), y) \right].
> $$
>
> In the above process, we focus on the parameter updates of $E$, which are dominated by gradients:
>
> $$
> \Delta E = -\eta \cdot \mathbb{E}_{(x,y) \sim \mathbb{P}_T} \left[ \frac{\partial \mathcal{L}\_{CE}(C(E(x)), y)}{\partial E}
> \right].
> $$
>
> Specifically, for the newly added class $\Delta \mathcal{Y}$, its corresponding samples may account for a large proportion of the loss gradient. Since the model had not encountered these classes initially, the prediction deviation is greater, resulting in a higher loss and steeper gradient. Hence, it will cause $E$ to be more inclined to update towards the new class, thereby leading to significant changes in the extracted feature values.
>
> In summary, we can conclude that input distribution shift (either from new classes or new domains) leads to larger $Diff^F$.
>
> ## **3. Analysis of Discovery 2 ($Diff^C$)**
>
> ### **3.1 Definition**
>
> $$
> Diff^C = ||C_T - C_S||_2 = \sqrt{ \sum _{i} ||w_i^{(T)} - w_i^{(S)}||_2^2 }.
> $$
>
> ### **3.2 Domain/Class increment**
>
> - Assume $C_S \in \mathbb{R}^{d \times K}$, $C_T \in \mathbb{R}^{d \times (K + \Delta K)}$
> - New class heads $w_j^{(T)}$ are initialized at zero and updated with large gradients due to untrained logits
> - Let $\mu = \mathbb{E}[||w_j^{(T)}||_2]$ for $j \in \Delta K$
> - Inputs change but label supervision remains stable, classifier fine-tuned slightly around $w_i^{(S)}$:
>
> $$
> ||w_i^{(T)} - w_i^{(S)}||_2 \approx \varepsilon,\quad \forall i \in \{1,...,K\}.
> $$
>
> #### **Class increment case**
>
> $$
> {Diff^C_{\text{class}}}^2 = \sum _{j=1}^K ||w_j^{(T)} - w_j^{(S)}||_2^2 + \sum _{j=K+1}^{K+\Delta K} ||w_j^{(T)}||_2^2 \approx K \cdot \varepsilon^2 + \Delta K \cdot \mu^2.
> $$
>
> #### **Domain increment case**
>
> $$
> {Diff^C_{\text{domain}}}^2 \approx K \cdot \varepsilon^2.
> $$
>
> #### **Comparative Inequality**
>
> Given $\mu \gg \varepsilon$:
>
> $$
> Diff^C_{\text{class}}  \gg  Diff^C_{\text{domain}}.
> $$
>
> This inequality theoretically confirms that class increments yield substantially higher classifier parameter variation, justifying our use of $Diff^C$ to detect new class knowledge.

---

> > ### Author Response · Authors · 2025-08-09
> >
> > Dear reviewer:
> >
> > Thank you again for taking the time to review our paper! As the discussion period is nearing its end, we want to ensure we have addressed all your concerns. If there are any additional points or feedback you'd like us to consider, feel free to let us know. Your insights are invaluable to us, and we're eager to address any remaining issues to improve our work. Thanks a lot!

---

### Official Review · Reviewer_7wkT · 2025-07-01

**Clarity:** 2
**Significance:** 3
**Originality:** 3
**Rating:** 4
**Confidence:** 3

**Summary:**

This work proposes a fine-grained federated domain adaptation method named Gains, designed to address the challenges of knowledge discovery and knowledge adaptation in open-set scenarios. By splitting the model into an encoder and a classifier, Gains is able to separately handle domain and class increments, and it utilizes a contribution-driven aggregation mechanism to achieve more efficient knowledge integration. Additionally, the paper introduces an anti-forgetting mechanism that prevents the degradation of source domain performance.

**Questions:**

1. The primary step of this work is detecting the type of increment. The manual setting of thresholds may lead to subjectivity in the results. The sensitivity of the DiffF and DiffC thresholds to model performance should be further explored.
2. In the experiments, are class increments and domain increments tested separately? Has the paper considered the case where both class and domain increments occur simultaneously in the target client? How would the aggregation process work in such a scenario?
3. The algorithm 1 overlooks one scenario: if the target client does not introduce new knowledge, the current framework will skip aggregating client's parameters, which is clearly thoughtless.
4. The experimental results in Table 1 lack explanations for two crucial observations: (1) Why do FOSDA and SemiFDA achieve T-Acc of 0.00 under Mild conditions but perform better in Medium and Strong cases? (2) Why do AutoFedGP and FedAVG exhibit S-ACC close to 0.00 in Mild scenarios?
5. The current study lacks sensitivity experiments on the balance coefficient $\lambda$, and an in-depth analysis is required.
6. It would be valuable to present baseline results under no domain shift conditions to more convincingly demonstrate the performance degradation phenomenon on the source domain as described in lines 37-38.
7. Gains requires computing multiple contributions and distance metrics. Does this increase model complexity? How does its training time compare with other methods?

**Ethical Concerns:**

["NO or VERY MINOR ethics concerns only"]

**Final Justification:**

The authors have responded thoughtfully to my concerns. In particular, their explanation on the sensitivity analysis of important hyper-parameters and experimental results on mixed incremental scenarios, were appreciated. Based on these improvements, I have decided to increase my score.

**Limitations:**

yes

**Quality:**

3

**Strengths And Weaknesses:**

Strengths

1. The fine-grained knowledge discovery method (distinguishing between domain and class increments) offers a novel solution to domain adaptation problems in federated learning. Compared to existing methods, Gains provides a more detailed handling of knowledge increments in open-set environments.
2. The anti-forgetting mechanism ensures that the integration of new knowledge does not degrade the performance of the source domain, thus maintaining stability throughout the adaptation process.
3. The experiments cover various levels of data shifts (mild, medium, and strong), validating the effectiveness of Gains across different scenarios.

Weaknesses

1. The work relies on manually set thresholds (such as DiffF and DiffC). While this approach works effectively in the experiments, it remains a challenge to automatically adjust these thresholds in response to continuously changing data in practical applications.
2. The current method seems to design separate aggregation strategies for class and domain increments. However, in real-world scenarios, class and domain increments may occur simultaneously. The paper does not fully address how to handle such mixed scenarios in practice.
3. The experimental section does not sufficiently support the paper's arguments, and more analysis of experimental results is needed. For example, there is a lack of hyperparameter experiments on the balance coefficient $\lambda$ in the Anti-forgetting mechanism.

---

> ### Author Rebuttal · Authors · 2025-07-30
>
> Thank you sincerely for your valuable comments. We have made every effort to respond to each point individually, addressing your concerns to the best of our ability. We hope these revisions adequately address all the issues you raised. Your input is greatly appreciated.
>
> ## Q1 & W1. Sensitivity analysis of the $Diff^F$ and $Diff^C$ thresholds
>
> Although the thresholds are manually set, the model exhibits strong robustness to threshold variations. As can be seen from Figure 2, the changes in $Diff^F$ and $Diff^C$ are very significant, which means that the thresholds can take values over a wide range, with $Diff^F$ ranging from 700 to 3400 and $Diff^C$ from 0.05 to 0.27. We also conduct experiment tests on various thresholds using the mild shift scenario in the DigitFive dataset as an example. Assuming the source domain data is MNIST-M and the target domain data is MNIST, with $T_F \in {800, 1000, 1200}$ and $T_C \in {0.20, 0.25, 0.27}$. The experimental results obtained are shown in the following table:
>
> | $Diff^F$ | T-Acc | S-Acc | G-Acc |
> |------|--------|--------|--------|
> | 800  | 99.62  | 92.01  | 93.15  |
> | 1000 | 99.34  | 93.21  | 94.44  |
> | 1200 | 99.24  | 93.06  | 93.91  |
>
> | $Diff^C$ | T-Acc | S-Acc | G-Acc |
> |-------|-------|-------|-------|
> | 0.20  | 99.75 | 92.34 | 93.29 |
> | 0.25  | 99.34 | 93.21 | 94.44 |
> | 0.27  | 99.86 | 92.71 | 93.01 |
>
> From the above two tables, it can be seen that the model performance remains stable when parameters fluctuate within reasonable ranges (performance variation < 1%).
>
> ## Q2 & W2. Mixed incremental scenarios
>
> When both domain shift and class shift occur simultaneously in the target domain, we can combine Domain-incremental and Class-incremental contribution-driven aggregation strategies. The aggregation strategy is as follows:
> - **Encoder**: Contribution calculated by Eq. (2), aggregated by Eq. (4).
> - **Classifier**:
>   - Old-class channels aggregated via Eq. (3) and Eq. (5),
>   - New-class channels are directly adopted from the target domain model, as described in Line 176 of the manuscript.
>
> To this end, we validated the scenario where both class increment and domain increment occur simultaneously. Specifically, using the DigitFive dataset medium shift as an example, the source domain client data is MNIST with classes {0,2,3,4,6,7,8,9}; the target client includes two types of data: one is MNIST classes {1,5}, and the other is SVHN dataset classes {0,2,3,4,6,7,8,9}. The experimental results obtained are as follows:
>
> | Method | T-Acc| S-Acc| G-Acc|
> |--------|------|------|------|
> | Gains-domain+class increment | 97.53 | 91.88 | 92.91 |
> | Gains-only domain increment | 97.91 | 90.09 | 91.65 |
> | Gains-only class increment | 99.34 | 93.21 | 94.44 |
>
> These results indicate that under the mixed incremental scenario, the performance achieved by Gains is comparable to scenarios with only domain or class increments, without significant degradation. We will include this new experiment and clarify the support for mixed-increment scenarios in the final version.
>
> ## Q3. Algorithm 1
>
> When no new knowledge is introduced, the original model can be directly applied to newly joined clients for inference tasks without requiring training. We will complete this explanation in the final version of the algorithm.
>
> ## Q4. Explanations of Table 1
>
> ### (1) The T-Acc of FOSDA and SemiFDA
>
> FOSDA is designed for open-set scenarios by leveraging an uncertainty-aware mechanism to distinguish known from unknown classes. Under a mild domain shift, the inter-domain discrepancy is too small to activate the unknown class detector properly, causing a large number of target samples to be incorrectly rejected, which leads to T-Acc approaching zero. As the domain shift increases (Medium/Strong), the feature distribution discrepancy becomes more pronounced, enabling FOSDA to perform effective domain separation.
>
> SemiFDA employs an unsupervised feature alignment strategy (based on CORAL) with a frozen classification head. In the Mild setting, the target distribution is extremely close to the source, resulting in insufficient feature updates and poor adaptation of the classifier—hence a T-Acc of 0.00. As the domain gap widens, stronger distributional shifts provide more adaptation signals, improving alignment and classification performance.
>
> ### (2) The S-Acc of AutoFedGP and FedAVG
>
> In the Mild scenario, AutoFedGP suffers from scarce and noisy target data. Its auto-weighting mechanism overemphasizes the limited target gradients and suppresses source gradient contributions, leading to severe source forgetting.
>
> FedAVG assumes fully IID data and lacks any domain adaptation mechanism. Under Mild conditions, the introduction of new target-domain classes exacerbates data distribution skew across clients, resulting in unstable training dynamics (as shown in Fig. 4(a)). Consequently, FedAVG fails to balance performance between source and target domains. The reported results reflect only the final training round.
>
> ## Q5 & W3. Sensitivity analysis of $\lambda$
>
> We test the performance on the DigitFive dataset under the medium shift scenario (task: USPS → MNIST) with $\lambda$ ∈ {0.0, 0.1, 0.3, 0.5, 0.7, 1.0}. Each $\lambda$ is tested with three repeated training runs, taking the average. The results are as follows:
> | λ   | T-Acc  | S-Acc  | G-Acc  |
> |-----|---------|---------|---------|
> | 0.0 | 99.20   | 83.35   | 85.87   |
> | 0.3 | 99.14   | 93.87   | 95.41   |
> | 0.5 | 98.99   | 99.41   | 99.34   |
> | 0.7 | 93.76   | 99.35   | 97.59   |
> | 1.0 | 85.55   | 99.43   | 93.47   |
>
> From the above experimental results, it can be seen that when $\lambda$ approaches 0, new knowledge dominates aggregation, leading to old knowledge forgetting (S-Acc decreases). When $\lambda$ is too large, target domain performance is limited (T-Acc decreases). Overall, $\lambda=0.5$ achieves optimal balance, and subsequent experiments are set accordingly.
>
> ## Q6. Results under no domain shift conditions.
>
> In the table below, we present the accuracy of the source domain clients without the entry of new clients. Compare with Table 1 of the paper, it can be seen that almost all baselines show a decline in performance on the source domain, while Gains has the smallest decline. We will include these results in Table 1 in the final version.
>
> | Scenario |DigitFive| Amazon Review|
> |----------|--------|------|
> |  Mild    |99.46 |-|
> |  Medium  |89.23 |81.70 |
> |  Strong  |94.32 |85.65 |
>
> ## Q7. Concerns about the training time.
>
> Inevitably, extra computational costs occur when calculating the contributions on the server. However, by calculating the weights based on contribution, more efficient aggregation can be achieved, thereby significantly reducing the number of federated iterations. Taking the DigitFive dataset in the mild shift scenario as an example, the consumed computing resources and the number of iterations are as shown in the table below:
>
> | Method           |  Rounds to converge | Convergence time |
> | ---------------- | ------------------ | ------------------------- |
> | **Gains (Ours)**     |  **5**                 | **807.45**                   |
> | FedHEAL          | 40                | 1368.4                    |
> | FedAVG           | 20                | 1977.20                   |
> | FedProx          | 40               | 6880.80                   |
> | FedProto         | 32                | 9519.68                   |
>
> Although Gains consumes more computational resources per round, it reduces the number of communication rounds by approximately 84.8%, resulting in lower overall computation costs. Moreover, since the contribution calculation is performed on the server, it does not impose an additional computational burden on the clients. On the contrary, the significantly reduced number of rounds substantially lowers the communication overhead for clients. The above results will be shown in the final version.
>
> Thank you for your time and efforts. We hope this has addressed your concerns and answered your questions.
> Please don’t hesitate to reach out if you have any further questions.

---

> > ### Comment · Reviewer_7wkT · 2025-08-04
> > **Thanks for the response**
> >
> > Thank you for the detailed response, the author effectively addressed my issue, I will increase my grade. Hope you can add these details to your appendix.

---

> > > ### Author Response · Authors · 2025-08-05
> > >
> > > Thank you greatly for your valuable time and comments. We will include this content in the appendix.

---

### Official Review · Reviewer_JgVT · 2025-07-02

**Clarity:** 3
**Significance:** 3
**Originality:** 3
**Rating:** 5
**Confidence:** 4

**Summary:**

This paper proposes a fine-grained federated domain adaptation (FDA) method for open-set scenarios where new clients dynamically join federated learning. It addresses two core challenges: Identifying whether new clients introduce new classes or new domains, i.e., Knowledge discovery; and integrating new knowledge while preserving source-domain performance, i.e., Knowledge adaptation. Experiments on DigitFive and Amazon Review datasets show superior accuracy and faster convergence over baselines under mild/medium/strong data shifts.

**Questions:**

(1) In line 135, "local training \hat{I} times", isn't the I defined in the previous text the total number of federated training rounds? Why is it the number of local training rounds here?

(2) In line 130, the superscript of encoder E should be capital S.

(3) Please explain how public datasets are constructed and add a discussion on alternatives to public datasets in the discussion section.

**Ethical Concerns:**

["NO or VERY MINOR ethics concerns only"]

**Final Justification:**

The authors have responded thoughtfully to my concerns. In particular, their clarification on computational complexity, which is critical in the context of federated learning, was appreciated. Based on these improvements, I have decided to increase my score.

**Limitations:**

Yes

**Quality:**

3

**Strengths And Weaknesses:**

(1) Strengths:

The paper presents a methodologically contribution to federated domain adaptation, distinguished by its novel fine-grained knowledge discovery framework that effectively discriminates between class and domain increments.

The work flows logically from empirical motivation to algorithmic design, with exceptionally thorough experimental validation across various data-shift scenarios and datasets, demonstrating SOTA results.

The inclusion of scalability tests, generalization analyses (20+ domain pairs), and ablation studies further solidifies its credibility, while public code release enhances reproducibility. Overall, I think it is well-organized and written.

(2) Weaknesses:

Some notations are confusing, such as the definition of "local training \hat{I} times" in Line 135, which may lead readers to misunderstand the relationship between local training rounds and federated training rounds.

Additionally, the superscript of encoder E in Line 130 should be capital S. The method for constructing public datasets and alternative approaches to public datasets are not adequately explained.

Further discussion on these aspects is needed in the discussion section. The computing complexity analysis is expected, especially when more clients participate.

Figure 3 is not cited in the text. In Figure 3, defined symbols such as E^S, C^S, E^T, and C^T can be used for abbreviation. There are too many font types and too many font size gradations, making it somewhat difficult to read.

---

> ### Author Rebuttal · Authors · 2025-07-30
>
> Greatly thank the reviewer for your time and valuable feedback. We have made every effort to solve your problem and sincerely hope to meet your requirements. The responses are as follows:
>
> ## W1 & Q1. Notations
>
> In the paper, $\hat{I}$ refers to the number of local training iterations performed on the new client after joining the federated learning. The updated model is then uploaded to the server for fine-grained identification of new knowledge types. If new knowledge is detected, global federated training is conducted for $I$ rounds. To avoid confusion, we will denote this local training epoch as $Q$ in the final version.
>
> ## W2 & Q2 & Q3. Superscript error \& Concerns about the public dataset
>
> (1) Superscript error
>
> Yes, the superscript of encoder E should be capital S. Thank you for your careful reading. We will correct the superscript of Encoder E in line 130 in the final version.
>
> (2) Public dataset
>
> In our experiments, the public data is sampled from the DigitFive and Amazon Review datasets, which are strictly disjoint from client data and only stored on the server. Generally speaking, it can be accessed from other datasets that are totally different from clients' data, such as Office-Home, thereby avoiding direct privacy leakage.
>
> To address scenarios without public datasets, a lightweight client-encoder distillation mechanism can be adopted. Specifically, the server can collect statistical summaries (e.g., encoder outputs’ mean and covariance) from clients instead of raw data, ensuring privacy. By comparing these anonymized feature statistics, the server can approximate $Diff^F$ and $Diff^C$ for knowledge discovery.
>
> ## W3. Computing complexity analysis
>
> Compared with traditional federated learning, Gains mainly increases the computational load during the server-side contribution calculation. Its complexity is 𝑂(N⋅P⋅d), where N is the number of source domain clients, P is the size of the public dataset, and d is the number of model parameters. Inevitably, extra computational costs occur during the above process. However, by calculating the weights based on contribution, more efficient aggregation can be achieved, thereby significantly reducing the number of federated iterations and reducing the overall training time. Taking the DigitFive dataset in the mild shift scenario as an example, the consumed computing resources and the number of iterations are as shown in the table below:
>
> | Method           | Rounds to converge    | Convergence time |
> | ---------------- | --------------------- | ---------------- |
> | **Gains (Ours)**     | **5**            | **807.45**           |
> | FedHEAL          | 40                    | 1368.4           |
> | FedAVG           | 20                    | 1977.20          |
> | FedProx          | 40                    | 6880.80          |
> | FedProto         | 32                    | 9519.68          |
>
> Among the methods with comparable learning effects, although Gains consumes more computing resources in each round, it reduces the number of iterations by 84.8%, thereby reducing the overall training time. We will show this result in the experiment section.
>
> ## W4. Concerns about Fig.3
> Sorry, it is a mistake. Figure 3 should be cited in line 117:
>
> "Framework. Inspired by the above empirical discoveries, we propose a fine-grained federated domain adaptation framework in open set, Gains (as shown in Fig. 3). Specifically, it consists of two main components: knowledge discovery and knowledge adaptation."
>
> In the final version, we will simplify the characters in Figure 3 to make them clearer and easier to understand.

---

> > ### Comment · Reviewer_JgVT · 2025-08-04
> > **Thanks for the response**
> >
> > Thank the authors for addressing my main concerns, particularly the clarification regarding computational complexity. These explanations significantly enhance the readability and practical relevance of the paper. I am satisfied with the revised version and am therefore willing to raise my original score accordingly.

---

### Official Review · Reviewer_va2g · 2025-07-02

**Clarity:** 3
**Significance:** 3
**Originality:** 3
**Rating:** 5
**Confidence:** 4

**Summary:**

This paper introduces a novel fine-grained federated domain adaptation framework in open set, named Gains, which aims to tackle the critical challenges of fine-grained knowledge discovery and rapid, balanced knowledge adaptation in federated learning. Specifically, Gains develops fine-grained knowledge discovery techniques to identify whether the new client introduces new knowledge and further discriminate its type (domain increment or class increment). Based on this, the model employs a contribution-driven aggregation strategy to prioritize source clients with higher contributions during the knowledge adaptation phase, accelerating the integration of new knowledge into the global model. Additionally, an anti-forgetting mechanism is incorporated to prevent performance degradation on the source domain, ensuring a balanced adaptation between the target and source domains. Extensive experiments conducted on multi-domain datasets across various data shift scenarios demonstrate that Gains significantly outperforms other methods in terms of performance for both source-domain and target-domain clients.

**Questions:**

-In Figure 2, it seems that the Encoder has the greatest Diff change in the new class. Why is it said that “the variation of the encoder does not show a clear fluctuation trend no matter what...”?
-In line 22, does "new clients, i.e., target domain" refer to new clients in the target domain or new clients from the target domain?

**Ethical Concerns:**

["NO or VERY MINOR ethics concerns only"]

**Final Justification:**

The authors' rebuttal has thoroughly addressed all my initial concerns, especially regarding large-scale experiments, formula clarity, and discussions. I'm convinced by their detailed clarifications and additional validations, significantly strengthening the paper's merit. I think it is a good work. Hence, as promised, I am delighted to upgrade the score to 5 accordingly.

**Limitations:**

-Equations (2) and (3) are somewhat perplexing and require additional descriptions of the formulas.
-According to the paper's description, Gains can reduce the number of federated iterations, thereby achieving rapid adaptation. However, this point is insufficiently emphasized in the method section of the paper.
-The discussion is not thorough enough. Future improvement directions should be provided in the discussion part.
-More experimental validations on larger-scale clients are expected.

**Quality:**

3

**Strengths And Weaknesses:**

strengths and weaknesses
The writing of this paper is clear and the topic is interesting. I will summarize the merits and flaws of this work based on my understanding. In terms of merits, I think this work is advantageous in three points:

(1) Innovative Research Approach. The paper proposes a fine-grained federated domain adaptation framework (Gains). It leverages the sensitivity of the encoder to domain shifts and the classifier to class increments, enabling fine-grained knowledge discovery and classification of knowledge types (domain increment or class increment). This approach is innovative and provides a new perspective for knowledge discovery and adaptation in federated learning.

(2) The experiments are extensive and convincing. The paper conducts extensive experiments to verify the effectiveness of Gains. In scenarios involving class increment and domain increment, Gains significantly outperforms other methods in target domain accuracy, source domain accuracy, and global accuracy. The paper also evaluates adaptation speed, generalization ability, and sequential federated domain adaptation scenarios.

(3) The code and data are open-sourced, providing a reference for future research and enabling other researchers to reproduce and build upon the work.

Though there exist no major concerns, I have a few comments as well as suggestions to improve this manuscript. These include:

(1) Lack of experimental validation in large-scale client scenarios. The current experiments only validated the effectiveness of Gains in a scenario with 50 clients (Section 4.1), but did not test scenarios with a larger scale of clients (e.g. 100+).It is recommended to supplement the experiments in two aspects: one is to expand the source-domain client number to 100; the other is to increase the number of clients carrying new knowledge in each batch during the sequential domain adaptation process from 1 to 10.

(2) Some formulas in the paper remain perplexing. For instance, it's hard to comprehend the physical meanings of formula (2) and formula (3).I believe the expressions of these formulas need to be simplified, or an explanation should be provided for each part of them.

(3) The discussion part is somewhat weak. In view of the problems existing in the current manuscript, what potential solutions could be explored in the future? To be specific, what paths for automated knowledge discovery could be investigated? And what are the scenarios of task - increment?

(4) The description of the effectiveness of the Gains method is somewhat vague. The paper repeatedly claims that Gains can enable the global model to quickly achieve federated domain adaptation. According to the experimental validation results presented later on (Line 273-281), this rapid adaptation stems from a reduction in the number of federated iterations, rather than the generally assumed computational cost. Therefore, I think it is necessary to clarify this in the methods section, or otherwise, it might lead to misunderstandings.

(5) Some typos can be found in the manuscript. For example, it seems that some of the separating lines in Table 2 and Table 3 have disappeared.

I would further update my score if my concerns are well addressed during the rebuttal.

---

> ### Author Rebuttal · Authors · 2025-07-30
>
> Thank you sincerely for your thoughtful comments and valuable insights on our manuscript. We have carefully addressed each of your concerns, dedicating significant effort to revise and clarify the points you raised. The following is our response:
>
> ## W1. Large-scale experiments
>
> Taking the medium shift in the DigitFive dataset (source: SVHN; target: MNIST) as an example, we respectively increase the number of source domain clients from 50 to 100, and increase the number of newly joined clients each time from 1 to 10, obtaining the following experimental results:
>
> | Method | T-Acc| S-Acc| G-Acc|
> | ------ | ---- | ---- | ---- |
> | Gains-100clients-1newclient | 93.09 | 92.99 | 93.02 |
> | Gains-50clients-1newclient | 97.91 | 90.09 | 91.65 |
> | Gains-50clients-10newclient | 95.67 | 87.29 | 90.35 |
>
> From the above table, it can be seen that when the number of source domain and target domain clients increases, the method proposed in the paper still maintains a good performance. This part will be included in the final version.
>
>
> ## W2. Explanations of formulas
>
> Formulas (2) and (3) are the foundation of our contribution-driven aggregation mechanism, designed to adaptively weigh each source client’s encoder and classifier during aggregation, depending on both their similarity to the target domain and their sample size.
>
> Formula (2) is as follows:
>
> $CD^E_n(i) = \frac{1}{(1 + \mathrm{Diff}^F_n(i)) \times \sum_{n=1}^{N} \left( \frac{1}{1 + \mathrm{Diff}^F_n(i)} \right)} \times \frac{\sum_{n=1}^{N} |\mathcal{D}^S_n|}{|\mathcal{D}^T| + \sum_{n=1}^{N} |\mathcal{D}^S_n|}$,
>
> where $\frac{1}{1 + Diff^F_{n}(i)}$ assigns a higher contribution to source clients whose encoder features are more similar to the target domain (i.e., with a smaller $Diff^F_n(i)$); the normalization $\frac{1}{\sum_{n=1}^{N} \frac{1}{1 + Diff^F_{n}(i)} }$ ensures fairness among all clients; the sample proportion component $\frac{\sum_{n=1}^{N} |\mathcal{D}^S_n|}{|\mathcal{D}^T| + \sum_{n=1}^{N} |\mathcal{D}^S_n|}$ ensures that source clients with more data have an appropriate level of influence. Formula (3) follows a similar logic but focuses on the classifier parameters ($Diff^C_n(i)$). A lower distance indicates a better fit to the target domain; thus, a higher contribution weight is assigned. We apologize for the inconvenience and will include explanations in the final version.
>
> ## W3. Concerns about the discussion
> ### (1) Solutions to automated knowledge discovery
>
> To automate the selection of thresholds, we can use a statistics-based approach. In the pre-experiment, we repeatedly introduce clients that carry new knowledge into the federated-learning system and, each time, record the resulting $Diff^F$ and $Diff^C$ values. After collecting many such samples, we compute the means ($\mu_F$, $\mu_C$) and standard deviations ($\sigma_F$, $\sigma_C$) of the two metrics. We then set the thresholds as:
>
> - $T_F = \mu_F + \alpha\sigma_F$,
> - $T_C = \mu_C + \beta\sigma_C$,
>
> where $\alpha$ and $\beta$ are tunable coefficients that control how strict the thresholds should be.
>
> ### (2) Solutions to complex increment scenarios
>
> When both domain shift and class shift occur simultaneously in the target domain, our method still works. We can combine Domain-incremental and Class-incremental contribution-driven aggregation strategies. The aggregation strategy is as follows:
> - **Encoder**: Contribution calculated by Eq. (2), aggregated by Eq. (4).
> - **Classifier**:
>   - Old-class channels aggregated via Eq. (3) and Eq. (5),
>   - New-class channels are directly adopted from the target domain model, as described in Line 176 of the manuscript.
>
> To this end, we validated the scenario where both class increment and domain increment occur simultaneously. Specifically, using the DigitFive dataset medium shift as an example, the source domain client data is MNIST with classes {0,2,3,4,6,7,8,9}; the target client includes two types of data: one is MNIST classes {1,5}, and the other is SVHN dataset classes {0,2,3,4,6,7,8,9}. The experimental results obtained are as follows:
>
> | Method | T-Acc| S-Acc| G-Acc|
> |--------|------|------|------|
> | Gains-domain+class increment | 97.53 | 91.88 | 92.91 |
> | Gains-only domain increment | 97.91 | 90.09 | 91.65 |
> | Gains-only class increment | 99.34 | 93.21 | 94.44 |
>
> These results indicate that under the mixed incremental scenario, the performance achieved by Gains is comparable to scenarios with only domain or class increments, without significant degradation. For more complex task-increment scenarios, such as extending from image classification tasks to object detection tasks, adaptive architectures can be designed by incorporating meta-learning techniques. This allows the model to autonomously identify and handle task space expansion, automatically differentiate parameters, and adapt independently, thereby ensuring continual knowledge growth and effective control of unlearning.
>
> We will include the above experimental validation and discussion in the final version.
>
> ## W4. Concerns about method effect
> In fact, rapid adaptation is reflected not only in fewer iterations but also in shorter computation time. Inevitably, extra computational costs occur when calculating the contributions on the server. However, by calculating the weights based on contribution, more efficient aggregation can be achieved, thereby significantly reducing the number of federated iterations and reducing the overall training time. Taking the DigitFive dataset in the mild shift scenario as an example, the consumed computing resources and the number of iterations are as shown in the table below:
>
> | Method           |  Rounds to converge | Convergence time |
> | ---------------- | ------------------  | ---------------- |
> | **Gains**       |  **5**               | **807.45**                    |
> | FedHEAL          | 40                    | 1368.4                    |
> | FedAVG           | 20                    | 1977.20                   |
> | FedProx          | 40                    | 6880.80                   |
> | FedProto         | 32                    | 9519.68                   |
>
> Although Gains consumes more computational resources per round, it reduces the number of communication rounds by approximately 84.8%, resulting in lower overall training time. We will further emphasize this point in the paper.
>
> ## W5. Missing table lines
> Thank you for your correction. We will revise the formats of Table 2 and Table 3 in the final version.
>
> ## Q1. Doubts about Figure 2
>
> As shown in Figure 2, in the class-incremental scenario, when the number of new classes is 1, the change in the encoder parameters is significantly lower than in the case where the new client does not experience a shift; when the number of new classes is 4, the amount of change is comparable to that when the new client does not experience a shift. This change increases as the number of new classes increases. This indicates that the amount of change in the encoder is closely related to the number of new classes, but not obviously related to whether new class knowledge is introduced. In addition, in the domain-incremental scenario, the change in the encoder is also comparable to that without increment. Therefore, the variation of the encoder does not show a clear fluctuation trend.
>
> ## Q2. Doubts about Line 22
>
> “new clients, i.e., target domain” refers to new clients from the target domain. We usually consider the data carried by newly joined clients as belonging to the target domain, and the target domain may or may not be the same as the source domain.

---

> > ### Comment · Reviewer_va2g · 2025-08-05
> >
> > The authors' rebuttal has thoroughly addressed all my initial concerns, especially regarding large-scale experiments, formula clarity, and discussions. I'm convinced by their detailed clarifications and additional validations, significantly strengthening the paper's merit. I think it is a good work. Hence, as promised, I am delighted to upgrade the score to 5 accordingly.

---

> > > ### Author Response · Authors · 2025-08-08
> > >
> > > Dear reviewer:
> > > Thank you very much for your encouragement and your time. As the discussion time is coming to an end, we still haven't seen any changes on our page. Could we kindly ask if you have confirmed the updated scores? (As the updated scores will be hidden from us). We apologize for any inconvenience caused.

---

### Note · Authors · 2025-08-12

Dear reviewers, ACs, and SACs：

Thank you for your careful reviews and the time you have invested in this work. During the rebuttal, the reviewers’ concerns were mainly grouped into three areas:

First, regarding the validations under more complex scenarios, we have conducted extra experiments on a larger scale, more challenging incremental settings, with more complex models, and more baselines. The results confirm that our algorithm remains effective. Technically, we perform experiments in typical and some complicated cases with real-world deployment considerations.

Second, several reviewers care about hyperparameter choices and method components. We have therefore performed a sensitivity analysis of the balance parameter between old and new knowledge (α) and of the knowledge-discovery threshold. The method shows a certain-level robustness to the threshold, while α plays the expected role in controlling the trade-off between old and new knowledge. To address the limitations raised in the discussion, we further propose an automatic threshold-selection procedure and a strategy that works even when no public dataset is available. We also supplement the paper with both theoretical and empirical complexity analyses, demonstrating that our algorithm accelerates iteration and markedly reduces computational cost.

Third, for the clearer explanation of the fine-grained knowledge-discovery mechanism in the reviewer AsDt's review, we actively provided experimental evidence and referenced analogous approaches in highly-cited papers for support. Meanwhile, we also added a detailed mathematical analysis that formally justifies the soundness of the knowledge-discovery procedure.

We sincerely hope that all remaining doubts can be resolved, and we will include all additional analyses in the appendix.

Thank you once again for your constructive comments.

---

### Decision · Program_Chairs · 2025-09-17

**Decision:**

Accept (poster)

**Comment:**

This paper considers federated domain adaptation (FDA) in open-set settings with fine-grained knowledge discovery (distinguishing class vs. domain increments), which is acknowledged as a novel problem setting by all reviewers. The proposed algorithm employs a contribution-driven aggregation strategy to prioritize source clients with higher contributions during the knowledge adaptation phase, and an anti-forgetting mechanism is incorporated to prevent performance degradation on the source domain.

The reviewers have some common concerns regarding 1) Validations under more complex scenarios, 2) Computational complexity, 3) Choice of hyperparameters, 4) Lack of theoretical justification for the discrimination mechanism. During the rebuttal and author-reviewer discussion phases, the authors have supplement additional empirical results to address most concerns of Reviewers va2g, JgVT, 7wkT.

The remaining concerns are mainly around lack of theoretical backing to the algorithm design. I agree with Reviewer AsDt that the proposed algorithm lacks of theoretical justification or derivation. Both contribution-driven aggregation and anti-forgetting mechanism seem based on weighted combination of models, where the weighting schemes are manually designed. Is it possible to learn the weights or derive the weighting scheme theoretically? In addition, the authors adopt the normalized $\frac{1}{1+Diff_n}$ as the weight for client $n$, considering the Gibbs distribution (where $ Diff_n $ is viewed as the energy) might be a better way.